# Large differences of highly oxygenated organic molecules (HOMs) and low volatile species in SOA formed from ozonolysis of β-pinene and limonene

Dandan Liu[1#], Yun Zhang[2,3#], Shujun Zhong[1], Shuang Chen[1], Qiaorong Xie[1], Donghuan Zhang[1], Qiang Zhang[1], Wei Hu[1], Junjun Deng[1], Libin Wu[1], Chao Ma[1], Haijie Tong[4,5], Pingqing Fu[1]

[1]Institute of Surface-Earth System Science, School of Earth System Science, Tianjin University, Tianjin 300072, China
[2]Innovation Center of Pesticide Research, Department of Applied Chemistry, College of Science, China Agricultural University, Beijing 100193, China
[3]Institute of Chemistry, Johannes Gutenberg University, Mainz 55128, Germany
[4]Multiphase Chemistry Department, Max Planck Institute for Chemistry, Mainz 55128, Germany
[5]Institute of Surface Science, Helmholtz-Zentrum Hereon, Geesthacht 21502, Germany
# These authors contributed to this study equally.

*Correspondence to:* Haijie Tong (haijie.tong@hereon.de); Pingqing Fu (fupingqing@tju.edu.cn).

**Abstract.** Secondary organic aerosols (SOA) play a key role in climate change and public health. However, the oxidation state and volatility of SOA are still not well understood. Here, we investigated the highly oxygenated organic molecules (HOMs) in SOA formed from ozonolysis of β-pinene and limonene. Fourier transform ion cyclotron resonance mass spectrometry (FT-ICR MS) was used to characterize HOMs in aerosol filter samples, and a scanning mobility particle sizer (SMPS) was used to measure the concentration and size distribution of SOA particles. The relative abundance of HOMs (i.e., ratio of summed mass spectrometry peak intensity of HOMs to totally identified organic compounds) in limonene SOA was 14~20%, higher than in β-pinene SOA (3~13%), exhibiting different trends with increasing ozone concentrations. β-pinene oxidation-derived HOMs exhibit higher yield at high ozone concentration, accompanied by substantial formation of ultra-low-volatility organic compounds (ULVOCs). Limonene-oxidation-derived HOMs exhibit higher yield at moderate ozone concentrations, with semi-, low-, and extremely low-volatility organic compounds (SVOCs, LOVCs, and ELVOCs) play a major role. Combined experimental evidence and theoretical analysis indicate that oxygen-increasing-based peroxy radical chemistry is a plausible mechanism for the formation of oxygenated organic compounds with 10 carbon atoms. Our findings show that HOMs and low volatile species in β-pinene and limonene SOA are largely different. The ozone concentration-driven SOA formation and evolution mechanism for monoterpenes is suggested to be considered in future climate or exposure risk models, which may enable more accurate air quality prediction and management.

# 1 Introduction

Secondary organic aerosols (SOA) are key component of airborne particulate matter, which play a crucial role in air quality, climate, hydrological cycle, and public health (Laden et al., 2006; Cohen et al., 2017; Kourtchev et al., 2016; Jokinen et al., 2015; Tröstl et al., 2016; Perraud et al., 2012; Ramanathan et al., 2001; Noziere et al., 2015). The SOA formed from oxidation of biogenic source volatile organic compounds (BVOCs) such as monoterpenes ($C_{10}H_{16}$) contribute a significant fraction of mass to total aerosols (Kanakidou et al., 2005; Hallquist et al., 2009; Ehn et al., 2012; Fu et al., 2009). However, biogenic SOA

formation is a complex multiphase process (Griffin et al., 1999; Hallquist et al., 2009; Jokinen et al., 2014; Shrivastava et al., 2017; Chen et al., 2011; Liu et al., 2016; Wang et al., 2023). The formation mechanism of SOA especially biogenic SOA is still largely unknown. Deep insights into biogenic SOA formation and evolution at a molecular level will facilitate the clarification of SOA's impacts on climate change and human health (Hallquist et al., 2009; Shah et al., 2019).

β-pinene and limonene are typical and important biogenic precursors that are released approximately 30.3 Tg yr$^{-1}$ globally

(Guenther et al., 2012). The molecular composition of these two compounds contains the same number of carbon atoms, hydrogen atoms, and double bond equivalents (DBE). However, β-pinene has a bicyclic structure with an exocyclic double bond, whereas limonene has a monocyclic structure with an exocyclic double bond and an endocyclic double bond, which is more reactive than the former one (Gallimore et al., 2017; Kenseth et al., 2018). Ozonolysis of limonene takes place on the endocyclic double bond (predominantly) as well as on the exocyclic double bond, while endocyclic double bond is the only

reactive site for β-pinene oxidation. Therefore, limonene exhibits greater reactivity toward ozone, resulting in a higher SOA yield than β-pinene (Tomaz et al., 2021; Bianchi et al., 2019). Laboratory simulations also confirmed that the molar yield of HOMs by β-pinene ozonolysis is much lower than that of limonene ozonolysis (Ehn et al., 2014; Jokinen et al., 2015). Thus, the structure-dependent reactivity of biogenic precursors is an important factor driving their role in atmospheric chemistry. Beyond this, the effects of different atmospheric oxidants on the SOA chemistry of biogenic precursors should also be clarified.

Gas phase ozonolysis, hydroxyl radical ($^{\bullet}$OH) chemistry, and nitrate radical ($NO_3$) oxidation etc. have been found as effective formation pathways for biogenic SOA (Kroll and Seinfeld, 2008; Kirkby et al., 2016), with the partition of low volatile organic compounds on exiting seed particles or homogeneous nucleation as key particle formation pathways (Donahue et al., 2012; Saukko et al., 2012). The relative contributions of different BVOCs oxidation pathways to atmospheric particulate matter pollution vary as the nature of parent VOC and atmospheric conditions (Mahilang et al., 2021). The yield of monoterpene SOA

also strongly dependents on oxidant types and concentrations. For instance, Zhao et al. found that the SOA yield of monoterpenes ozonolysis was higher than that of $^{\bullet}$OH oxidation (Zhao et al., 2015). Waring et al. also found that the number concentration of α-terpineol SOA was higher than limonene SOA at higher but not at lower concentrations of ozone (Waring et al., 2011). Moreover, the oxidant (e.g., $O_3$ or $^{\bullet}$OH)-dependent SOA yield difference of β-pinene has a different magnitude

from limonene (Jokinen et al., 2015; Mutzel et al., 2016). To unravel the underlying multiphase chemistry of these complicate

processes, the composition and volatility of biogenic SOA particles should be addressed.

Previous studies indicate that biogenic SOA comprise thousands of organic compounds, which exhibit a wide range of volatilities (Donahue et al., 2012; Ehn et al., 2014; Simon et al., 2020). Based on grouping the estimated effective saturation mass concentrations $C_0$ (Schervish and Donahue, 2020), the volatility of organic aerosols has been categorized into volatile organic compounds (VOCs, $C_0 > 3\times10^6$ µg m$^{-3}$), intermediate volatility OC (IVOCs, $300 < C_0 < 3\times10^6$ µg m$^{-3}$), semivolatile

OC (SVOCs, $0.3 < C_0 < 300$ µg m$^{-3}$), low-volatile OC (LVOCs, $3\times10^{-5} < C_0 < 0.3$ µg m$^{-3}$), extremely low-volatile OC (ELVOCs, $3\times10^{-9} < C_0 < 3\times10^{-5}$ µg m$^{-3}$), and ultralow-volatile OC (ULVOCs, $C_0 < 3\times10^{-9}$ µg m$^{-3}$), respectively (Hallquist et al., 2009; Simon et al., 2020). Iyer et al. suggested that after a single oxidant attack, BVOCs can be oxidized to low-volatility species on sub-second timescales, which consequently undergo decomposition or new particle formation (Iyer et al., 2021). Recent studies also showed that oxidation of BVOCs can produce large amounts of SOA particles via the nucleation of ULVOCs with the

absence of sulfuric acid (Kirkby et al., 2016; Guo et al., 2022). As a result, the irreversible distribution of (extremely) low volatile oligomer on the aerosol surface is expected to be enhanced (Zhang et al., 2017). Beyond this, ELVOCs have been found playing a crucial role in the generation of atmospheric cloud condensation nuclei (Kerminen et al., 2012) and new particle formation in most continental regions (Jokinen et al., 2015). To disclose the role of different VOC subgroups in the formation and environmental impacts of SOA, it is of critical importance to chemically resolve their oxidation state and linkage

with SOA evolution processes.

Highly oxygenated organic molecules (HOMs) have been found as a class of O-enriched multifunctional organic compounds (Tröstl et al., 2016; Zhang et al., 2017; Kirkby et al., 2016), which play an important role in the early growth of atmospheric organic aerosols (Ehn et al., 2014; Wang et al., 2020) and are closely associated with the formation of aqueous radicals (Tong et al., 2019). Ehn et al. found that HOMs in Hyytiälä's atmosphere, laboratory-generated α- and β-pinene SOA always have a

O/C $\geq$ 0.7 (Ehn et al., 2012). Tröstl et al. suggested that α-pinene SOA-contained HOMs can be defined as $C_xH_yO_z$ with x = 8~10, y = 12~16 and z $\geq$ 6 for monomer and $C_xH_yO_z$ with x = 17~20, y = 26~32 and z $\geq$ 8 for dimer (Tröstl et al., 2016). Tu et al. defined HOMs in fresh and aged biogenic (α-pinene, β-pinene, and limonene) SOA as assigned formulas having either O/C $\geq$ 0.6 or carbon oxidation states $OS_C \geq 0$, which were also categorized into highly oxygenated and highly oxidized HOMs (O/C $\geq$ 0.6 and $OS_C \geq 0$), highly oxygenated but less oxidized HOMs (O/C $\geq$ 0.6 but $OS_C \geq 0$), and highly oxidized HOMs

with a moderate level of oxygenation ($OS_C \geq 0$ but H/C < 1.2) for exploring the relative importance of oxygen content versus oxidation state (Kroll et al., 2011; Tu et al., 2016). Further study showed that monoterpene SOA-contained HOMs mainly composed of ELVOCs, LVOCs, and a small proportion of SVOCs (Li et al., 2019). Beyond the biogenic SOA, aromatic SOA and aged soot particles have also been found containing substantial fraction of HOMs (Molteni et al., 2018; Li et al., 2022). Respect to the formation mechanism of HOMs, autoxidation has been suggested to be an important pathway (Crounse et al.,

2013; Rissanen et al., 2014). For instance, peroxyl radicals ($RO_2$) can undergo an intramolecular hydrogen atom shift (H-shift) to form a hydroperoxide functionality (HOO-) and an alkyl radical (RO), and then molecular oxygen rapidly attaches to form a new more oxidized $RO_2$ radical, and be repeated several times to form HOMs (Bianchi et al., 2019). Given the different yield,

lifetime, and reactivity of HOMs in different types of SOA (Ehn et al., 2014; Jokinen et al., 2015; Pullinen et al., 2020; Shen et al., 2021; Guo et al., 2022), it is necessary to differentiate the compositional characteristics of HOMs in different types of SOA.

In short, BVOCs in the atmosphere undergo diverse oxidation chemistry pathways. Previous studies have shown the complicated synergy of BVOCs and atmospheric oxidants such as ozone in atmospheric processing (Rohr et al., 2003; Pathak et al., 2012; Wu et al., 2020). However, the compositional response of biogenic SOA especially the particulate HOMs is largely unknown. Given the continued even worsened ozone pollution scenarios widely (Li et al., 2023), a better understanding of the interconnections among ozone concentrations, BVOCs types, and HOM distributions will be extremely important. Moreover, laboratory studies often use high concentrations of ozone to reduce the loss of semi-volatile and low volatile vapors to the walls of the chamber (Pathak et al., 2008). Thus, deepened understanding of ozone concentration-dependent SOA and HOM formation mechanisms will also enable more accurate, reproducible, and reliable chamber studies.

The aim of this study was to analyze the influence of ozone concentration on the chemical composition of SOA formed from β-pinene and limonene ozonolysis. The experiments were carried out in a flow tube reactor at three different ozone concentrations. Then Fourier transform ion cyclotron resonance mass spectrometer equipped with a 7 Tesla superconducting magnet (7T FT-ICR MS) was used to study the molecular composition and formation mechanism of SOA. These efforts provide highly accurate molecular mass measurements of organic compounds to clearly assign formulas including carbon, hydrogen, and oxygen up to 850 Da, which enables the differentiation of particulate HOMs originating from β-pinene and limonene oxidation.

## 2 Method

### 2.1 Laboratory SOA generation and collection

A schematic description of the experimental procedure used in this study is shown in Figure 1. Laboratory SOA were generated by gas-phase ozonolysis of β-pinene or limonene in a 7 L quartz flow tube reactor. More detailed information about this reactor has been described in previous studies (Tong et al., 2016; Tong et al., 2019). Briefly, 1 mL of β-pinene (99%, Sigma Aldrich) or limonene (99%, Sigma Aldrich) were separately kept in 1.5 mL amber glass vials (VWR International GmbH) as SOA precursor sources. A flow of 1 bar and 0.1 standard liter per minute (slpm) $N_2$ (99.999%, Westfalen AG) was used to evaporate the volatile organic compounds (MFC2). Another 1 slpm $N_2$ flow was used as a diluting and carrier gas (MFC1) to introduce the gas phase precursors into the reactor for ~2.5 min gas-phase ozonolysis reaction. The $O_3$ was generated via passing synthetic air (Westfalen AG, 1.7 L min$^{-1}$) through MFC3 and a 185 nm UV light ($O_3$ generator, L.O.T.-Oriel GmbH & Co. KG). The ozone concentrations in the flow tube reactor were 50±10 ppb, 315±20 ppb or 565±20 ppb, which were measured with an ozone monitor (model 49i, Thermo Fisher Scientific Inc.). Based on a calibration function measured by gas chromatography-mass spectrometry, the precursor concentration was estimated to be in the range of 2~4 ppm for β-pinene

and 1~3 ppm limonene. The experimental conditions, aerosol concentrations, and collection efficiencies of this study are shown in Table S1. Ozonolysis of β-pinene and limonene SOA were performed under dark and dry conditions to reduce the complexity of SOA formation. Seed aerosols and hydroxyl radical scavenger were not added. When concentrations of β-pinene SOA, limonene SOA, and ozone are stable, the SOA were collected twice in a row for each ozone concentration condition on 47 mm diameter Omnipore Teflon filters (JVWP04700, Merck Chemicals GmbH). The sampling time varied from minutes to hours depending on the required aerosol mass. SOA filter samples were wrapped in aluminum foil and kept cool during transport between laboratories. A scanning mobility particle sizer (SMPS, GRIMM Aerosol Technik GmbH & Co. KG) was used to characterize the size distribution, number concentration, and mass concentration of the generated SOA. A flow rate of ~2.8 L min$^{-1}$ was controlled using a common diaphragm vacuum pump (0~3 L min$^{-1}$), which was connected with aerosol samplers. The condensation of water vapor on a filter during SOA collection were negligible in this study. A Teflon filter with particle loading was weighed using XSE105DU balance with accuracy of ±10 µg. It is noted that dilution induced an oxygen concentration drop in the flow tube. The impacts of oxygen concentration on HOMs formation and evolution are out of the interest of this study but warranty to be explored in follow up studies.

## 2.2 FT-ICR MS measurement

To measure water-soluble organic compounds, the β-pinene and limonene SOA filter samples and blank filters were extracted three times with ultrapure Milli-Q water. Each extraction was carried out in a sonicating ice bath for 10 min. The extracts were combined and added to a solid-phase extraction (SPE) cartridge (Oasis HLB, Waters Corporation, 60 mg, 3 mL) on the Supelco Visiprep SPE Vacuum Manifold (USA), which had been preconditioned with 3×3 mL methanol and Milli-Q water, respectively. Then, the cartridges were washed three times with 6 mL Millipore Q water and dried under a nitrogen flow for around 1 h. Subsequently, the organic compounds retained on the cartridge were eluted using 6 mL of methanol. The eluate was immediately concentrated to about 10 µL by a rotary evaporator and sample concentrator to optimize the concentration of SOA extracts. Low molecular weight compounds (< 100 Da) are expected to be excluded in the rinsing and drying steps of extraction, as reported by Bianco et al (Bianco et al., 2018). Finally, the eluate was stored at -20 °C in a brown glass vial with TEFLON® cap until analysis.

The chemical composition of analyte in pretreated extracts were finally analyzed with a 7 T FT-ICR MS (Bruker Daltonik, GmbH, Bremen, Germany) equipped with an electrospray ionization (ESI) ion source at the School of Earth System Science, Tianjin University, Tianjin, China. The instrument was externally calibrated in the negative ion mode with Suwannee River fulvic acid (SRFC) standard and the resulting mass accuracy was better than 1 ppm. All the extract samples were infused into the ESI unit by syringe infusion at a flow rate of 220 µL h$^{-1}$ and analyzed in negative ionization mode. Ions were accumulated for 0.05 s in the hexapole collision cell. Each mass spectra ranged from 150 to 1000 Da. The ESI capillary voltage was 5.0 kV and the spectra are based on the accumulation of 256 scans. An average resolving power (m/Δm 50 %) of over 400 000 (at mass-to-charge (m/z) 400 Da) was achieved. The capillary temperature was maintained at 200 °C. Filter blank was analyzed

following the same procedure as the aerosol samples. Other details of the experiment setup can be found elsewhere (Cao et al., 2016; Xie et al., 2020a).

## 2.3 Molecular formula assignment

The original FT-ICR MS data was processed using Data Analysis 5.0 (Bruker Daltonics). The mass spectra were internally
recalibrated using an abundant homologous series of oxygen-containing organic compounds in the samples over the mass range between 150 and 1000 Da. Molecular formulas were assigned for peaks with a signal-to-noise (S/N) ratio $\geq 4$ by allowing a mass error threshold of $\pm 1$ ppm between the measured and theoretically calculated mass. The molecular formula calculator was set to calculate formulae in the mass range between 150 and 800 Da with elemental compositions up to 40 carbon (C), 80 hydrogen (H), and 30 oxygen (O) atoms with a tolerance of $\pm 1$ ppm when only C, H and O are studied. Furthermore, double
bond equivalents (DBE) must be an integer value, the elemental ratio limits of hydrogen-to-carbon ratio (H/C) (0.3~2.5), oxygen-to-carbon ratio (O/C) (0~1.2), and a nitrogen rule for even electron ions were used to eliminate chemically unreasonable formula (Koch et al., 2005). Unambiguous molecular formula assignment was determined with help of the homologous series approach for improving the reliability on multiple formula assignments (Koch et al., 2007). No isotopic peaks were considered in this study, but more information on the FT-ICR-based analysis of isotopes in ambient aerosols can
be found from our previous study (Xie et al., 2022). All organic molecules with S/N $\geq 20$ and intensity greater than those of the analyzed samples were blank corrected. The lower peak intensity of common ions suggests that they were resulted from carry-over within the electrospray ionization source (Kundu et al., 2012). Because of the instrument limitation, the absolute mass concentration of each compound cannot be obtained. It is noted that peak intensity of MS spectrum is not directly translatable to abundance or concentration.
The assigned molecular formulae were examined using the DBE and Kendrick mass defect (KMD) series (Wu et al., 2004). To assess the saturation and oxidation degree of β-pinene and limonene SOA, the value of DBE is calculated along Eq. (1).

$$DBE = 1 + C - 0.5H ,\qquad\qquad (1)$$

where $C$ and $H$ are the number of carbon and hydrogen atoms, respectively.

The maximum carbonyl ratio (MCR) was used to estimate the contribution of carbonyl equivalent groups in the molecule with
oxygen number greater than or equal to DBE (Zhang et al., 2021). The value of MCR is calculated as Eq. (2).

$$MCR = \frac{DBE}{O} ,\qquad\qquad (2)$$

where $O$ is the number of oxygen atoms in the formula. Based on the MCR values, HOMs were categorized into 4 groups: (I) very highly oxidized organic compounds (VHOOCs; $0 \leq MCR \leq 0.2$), (II) highly oxidized organic compounds (HOOCs; $0.2 < MCR \leq 0.5$), (III) intermediately oxidized organic compounds (IOOCs; $0.5 < MCR \leq 0.9$), and (IV) oxidized unsaturated
organic compounds (OUOCs; $0.9 < MCR \leq 1$).

The carbon oxidation states ($OS_C$) is used to describe the composition of complex mixtures of organic matter undergoing oxidation processes. $OS_C$ is calculated as follows (Kroll et al., 2011):

$$OS_C = 2O/C - H/C \,, \tag{3}$$

The molecular weight (MW), O atom, O/C ratio, $OS_C$ and DBE was calculated using Eq. (4).

$$X = \sum(Int_i \times X_i)/\sum Int_i \,, \tag{4}$$

where $X$ is the mean value of different elemental characteristics ($X_i$), and $Int_i$ is the mass spectra peak intensity for each formula, $i$.

Molecular corridors can help to constrain chemical and physical properties as well as reaction rates and pathways involved in organic aerosol evolution (Shiraiwa et al., 2014). Saturation vapor pressure ($C_0$) is a consequence of the molecular characteristics of molar mass, chemical composition, and structure. Li et al. have developed a parameterization to estimate $C_0$ as $log_{10}C_0$ = f ($n_C$, $n_O$) (Li et al., 2016). The $C_0$ is defined by the 2D volatility basis set (2D-VBS) as follows,

$$log_{10}C_0 = (n_c^0 - n_c)b_C - n_O b_O - 2\frac{n_C n_O}{n_C - n_O}b_{CO} \,, \tag{5}$$

where $n_c^0$ is the reference carbon number; $n_C$ and $n_O$ denote the number of carbon and oxygen atoms, respectively; $b_C$ and $b_O$ denote the contribution of each kind of atoms to $log_{10}C_0$, respectively, and $b_{CO}$ is the carbon-oxygen nonideality (Donahue et al., 2011). The above parameterization method was adapted to the work of Li et al. (2016). The target compounds were categorized into intermediate volatility organic compounds (IVOCs, $300 < C_0 < 3\times10^6$ µg m$^{-3}$), semivolatile OC (SVOCs, $0.3 < C_0 < 300$ µg m$^{-3}$), low-volatile OC (LVOCs, $3\times10^{-5} < C_0 < 0.3$ µg m$^{-3}$), extremely low-volatile OC (ELVOCs, $3\times10^{-9} < C_0 < 3\times10^{-5}$ µg m$^{-3}$), and ultra-low-volatile OC (ULVOCs, $C_0 < 3\times10^{-9}$ µg m$^{-3}$), respectively (Donahue et al., 2011; Bianchi et al., 2019; Schervish and Donahue, 2020).

## 2.4 Determination of highly oxygenated molecules (HOMs)

Due to their low saturation vapor pressure, ambient HOMs (Vogel et al., 2016) and laboratory-generated HOMs (Jokinen et al., 2015; Roldin et al., 2019; Peräkylä et al., 2020) frequently comprise low-volatility organic compounds (LVOCs) even extremely low-volatility organic compounds (ELVOCs). In this study, HOMs with molecular formulae of $C_{8-10}H_{12-16}O_{6-9}$ and $C_{17-20}H_{26-32}O_{8-15}$ were assigned to HOM monomers and dimers formed from monoterpene ozonolysis, and compounds with O/C ratio < 0.7 was used to filter out non-HOM monomers (Tu et al., 2016; Tröstl et al., 2016; Tong et al., 2019). The formation pathways of HOMs were estimated based on previous research, and mainly through hydroperoxide channel and alkoxy channel (Tomaz et al., 2021; Shen et al., 2021; Kundu et al., 2012). It is noted that the current definition of HOMs is different from previous studies and does not count in HOM trimmers or other HOMs with higher oligomerization degrees, which is warranty to be explored in follow up studies.

# 3 Results and discussion

## 3.1 Effects of ozone concentration on size distribution and oxidation state of SOA

Figure 2 shows the averaged particle size distribution and number concentration of β-pinene and limonene SOA during the aerosol sampling period. Overall, limonene SOA exhibit a broader size distribution range than β-pinene SOA at lower $O_3$ concentrations (50 and 315 ppb), and the number-size distribution of limonene SOA at 315 ppb $O_3$ exhibits a shoulder peaks profile, which is different from the mono peak profile of β-pinene SOA. As the ozone concentration is increased from 50 to 565 ppb, the peak number concentration of β-pinene and limonene SOA increased for 5-fold (from 0.7 to $3.5 \times 10^6$ $cm^{-3}$) and 1.8-fold (from 2.2 to $3.9 \times 10^6$ $cm^{-3}$), respectively. Accordingly, the dominant size range of β-pinene and limonene SOA expanded from 10~80 to 10~200 nm and from 10~100 to 10~200 nm, and the size of particles with peak number concentrations shifted from both 32 nm to 70 and 80 nm, respectively. Based on the preassigned particle density of 1 g $cm^{-3}$, the ~100 nm diameter β-pinene SOA and ~123 nm diameter limonene SOA particles that formed at 565 ppb $O_3$ exhibit the highest mass concentration of ~1152 and ~1484 µg $m^{-3}$, respectively (Figure S1 in SI). The precursor-dependent number- and mass-size distribution profiles in Figures 2 and S1 may be related to different partition and agglomeration kinetics of low volatile organics, with the former process playing a plausible stronger role. The "partition" here means an equilibrium between the absorption and desorption rates of oxidized β-pinene or limonene products from SOA surfaces (Kamens et al., 1999), and a gas/particle partitioning absorption model has been found able to describe SOA yield well (Takekawa et al., 2003; Song et al., 2011). The higher SOA yield at higher ozone concentrations is associated with the accelerated condensation or gas-particle partition of low-volatile organics at higher ozone concentrations, which can promote the formation and growth of molecular clusters (Shrivastava et al., 2017). This claim is supported by previous findings that ozonolysis of limonene and monoterpenes exhibits high yield of extremely-low volatile organic compounds (ELVOCs) and can produce substantial amounts of SOA even at low ozone concentrations (Waring et al., 2011; Jokinen et al., 2015).

The molecular weight (MW), O atom number, O/C ratio, double bound equivalents (DBE), carbon oxidation state ($OS_C$), and VOC subgroups of β-pinene SOA and limonene SOA were shown in Table 1. It shows that particulate organics with higher $OS_C$, higher MW, and lower volatility exhibit larger contribution to SOA mass at higher ozone concentrations. High concentration of ozone tends to influence $RO_2$ levels and convert less oxidized organic molecules to highly oxygenated organic molecules (HOMs) via oxygen-increasing-reactions (OIR), e.g., producing new alkyl radicals through $O_2$ addition to an existing alkyl radical → reaction between $RO_2$ radicals → isomerization of the alkoxy radicals (Kundu et al., 2012). The overall higher fraction of ULVOCs in β-pinene SOA than limonene SOA is in line with previously observed higher abundance of organic peroxides and aqueous radical yield of β-pinene SOA (Badali et al., 2015; Tong et al., 2016; Tong et al., 2018). This reflects the importance of ozone concentration in determining the oxidative potentials of SOA. For limonene SOA, when the ozone concentration was increased from 315 to 565 ppb, the MW, O atom and DBE are decreased, indicating that high carbon- and oxygen-containing organic molecules in limonene SOA may fragment at high ozone concentration to form low carbon number and less oxidized organic molecules. The element characteristic values of limonene SOA were generally higher than β-pinene SOA, probably due to the following reasons. First, ozonolysis of limonene proceeds in a faster rate ($k_{\text{limonene+O}_3}$ = $2.1 \times 10^{-16}$ $cm^3$ molecules$^{-1}$ s$^{-1}$) than β-pinene ($k_{\text{β-pinene+O}_3}$ = $1.5 \times 10^{-17}$ $cm^3$ molecules$^{-1}$ s$^{-1}$) (Atkinson and Arey, 2003).

Second, limonene is more inclined to undergo oxygenate and accretion reactions than β-pinene. Third, non-condensation reactions that dominated by hemi-acetal reactions followed by hydrperoxide and Criegee radical reactions might play an important role in the limonene SOA formation (Kundu et al., 2012).

Figure S2a shows the summed MS spectra intensity of identified organics in β-pinene and limonene SOA samples, and the compounds with MW of 150~450 Da account for a major fraction. As the ozone concentration was increased from 50 to 315 and 565 ppb, the total spectra intensity of β-pinene SOA increased continually, whereas that of limonene increased first and then decreased. Figure S2b shows the formula number of identified organic molecules in SOA. The total formula number of β-pinene SOA increased first and then decreased as the increasing ozone concentrations, but the formular number of limonene SOA gradually decreased. The different trends from summed MS spectra intensity to formula number reflect the complexity of SOA mass and composition evolution, which varied as precursor types.

Table S2 shows that predominant molecules in β-pinene SOA are $C_{17}H_{26}O_4$ (m/z = 293) and $C_{10}H_{16}O_3$ (m/z = 183, pinonic acid) at 50 ppb ozone concentration (Jaoui and Kamens, 2003), and $C_{19}H_{30}O_5$ (m/z = 337), $C_{10}H_{16}O_3$ (m/z = 183), $C_9H_{14}O_4$ (m/z = 185), and $C_{19}H_{30}O_7$ (m/z = 369) at 315 and 565 ppb ozone concentrations. A higher relative abundance of $C_{32}H_{42}O_3$ (m/z = 473) was only found in β-pinene SOA formed at 50 ppb ozone condition, which is high carbon-containing and less oxygen-containing compound, indicating that β-pinene ozonolysis products prefer carbon-carbon bonding or oligomerizing at low ozone concentration. The mostly abundant organics in limonene SOA is always 7-hydroxy limononic acid ($C_{10}H_{16}O_4$, m/z = 199) at three ozone concentrations, which has also been observed as a major product from limonene ozonolysis in previous studies (Kundu et al., 2012; Gallimore et al., 2017; Hammes et al., 2019). Low oxygen-containing organic molecules (monomer: $O \leq 2$ and dimer: $O \leq 4$) exhibit a higher relative abundance in β-pinene and limonene SOA at lower ozone concentrations, indicating the preferred formation of high oxygen-containing organic molecules at higher concentrations of ozone, which may be associated with the indirectly accelerated autoxidation via $O_3$-augmented $RO_2$ levels etc.

For a better characterization of the formation and evolution processes of SOA, the maximum carbonyl ratio (MCR) index was used here (Zhang et al., 2021). Figure 3 shows the components of HOMs in β-pinene and limonene SOA depicted in an MCR-Van Krevelen (VK) diagram. Most of the plotted compounds are located in regions II and III, with the former region covers more compounds. Only a few compounds show up in regions I and IV. Such a distribution pattern indicates that most of the aerosolized oxidation products from ozonolysis of β-pinene and limonene are intermediately oxidized organic compounds (IOOCs) and oxidized unsaturated organic compounds (OUOCs) with MCR values of 0.5~1.0. Very little amount of organic compounds have MCR values < 0.2 or > 0.9, reflecting the low abundance of highly oxidized organic compounds (HOOCs) or highly unsaturated organic compounds (HUOCs). It is noted that MRC cannot differentiate the functional groups with the same formular, the improvement of which is warranty to be explored in follow up studies. Interestingly, the maximum relative abundance value of HOMs in β-pinene SOA elevated as the increasing ozone concentration, whereas that of limonene SOA decreased accordingly. Thus, high concentration of ozone might have inhibitory effect on the formation of limonene SOA-associated oxygenated organic compounds (e.g., due to ozonolysis-produced •OH radicals begin to cause more oxidative

fragmentation of first-generation products and condensing compounds (Hallquist et al., 2009; Kroll et al., 2007; Zhao et al., 2015), whereas an enhancement effect on the formation of β-pinene SOA-associated organic compounds.

**3.2 Composition and relative abundance of HOMs in β-pinene and limonene SOA**

As shown in Figure 4, the identified HOMs mainly exist with $m/z$ of 350~450 Da. The relative fractions of HOMs ($RA_{HOMs}$) in limonene SOA are higher than β-pinene SOA. As the ozone concentration was increased from 50 to 315 ppb, the MS spectra peak number-based $RA_{HOMs}$ in β-pinene and limonene SOA kept constant to be ~3% and ~5%, whereas the peak intensity-based $RA_{HOMs}$ increased significantly from 3% to 7% and 14% to 20%, respectively. This indicates that HOMs yield rather than chemical composition diversity of β-pinene and limonene SOA responded to the increasing ozone concentration. As the ozone concentration was increased further to 565 ppb, both peak number- and peak intensity-based $RA_{HOMs}$ in β-pinene SOA increased significantly to 5% and 13%, indicating a prominent change of HOMs yield and composition. In contrast, the peak number-based $RA_{HOMs}$ in limonene SOA still did not change, and the peak intensity-based $RA_{HOMs}$ even decreased slightly to 18%, reflecting the main change of HOMs yield. Previous studies found that addition of $O_3$ occurs mainly on the endocyclic double bond, which tends to generate ˙OH, and higher proportion of stabilized Criegee intermediates (SCIs) can be formed from the endocyclic double bond than from the exocyclic double bond (Wang and Wang, 2021; Gong et al., 2018). Therefore, we estimate that ˙OH produced by the ozonation of endocyclic double bond of limonene may compete with $O_3$ towards the formation and evolution of particulate HOMs (Atkinson et al., 1992; Kristensen et al., 2016).

Figure 5 shows the O/C, H/C, carbon number, and $OS_C$ of HOMs and non-HOM molecules in β-pinene SOA and limonene SOA samples. Green data points were found more than red data points, meaning that larger number of molecules formed by ozonolysis of β-pinene than limonene. The oxidation state distribution of unique molecules (i.e., compounds that were only found) in β-pinene SOA was broad (Figure 5a) and mostly are located in the low O/C ratio region of 0~0.4 (Figure 5b). In contrast, most of the unique molecules in limonene SOA are located in the high oxidation state region of $OS_C > -1.5$ (Figure 5a and 5b) and moderate O/C ratio region of 0.3~0.7 (Figure 5b). Such a profile difference indicates a higher ozonation degree of limonene than β-pinene. The HOMs (Figure 5c and 5d) show a similar variation trends as the SOA (Figure 5a and 5b), confirming the close association of HOMs with the formation of biogenic SOA. The substantial number of grey data points in Figure 5a-5d reflects the significant composition similarities between β-pinene and limonene SOA, including the HOMs.

The identified HOMs in β-pinene and limonene SOA were categorized into groups with different carbon number ($C_n$) or oligomer clusters, the relative fractions of which are plotted versus the ozone concentration in Figure 6. Figure 6a shows that HOM monomers are mostly enriched in β-pinene and limonene SOA that formed at 315 ppb ozone condition. Therein, the limonene SOA-contained HOM monomers mainly exist as $C_{10}$ species, while β-pinene SOA-contained HOM monomers mainly exist as $C_8$ species. This is due to carbon backbone of endocyclic limonene is retained on ozonolysis, whereas the terminal vinylic carbon of exocyclic β-pinene is cleaved (Ma and Marston, 2008; Kundu et al., 2012). Figure 6b shows that the relative abundance of dimers in β-pinene and limonene SOA exhibit different variation trends as the increasing ozone

concentrations. The carbon numbers of HOM dimers in β-pinene SOA exhibit the same pattern at 50 and 315 ppb ozone concentrations, with the relative abundance increased from $C_{17}$ to $C_{18}$ and then gradually decreased for $C_{19}$ and $C_{20}$ species. However, at 565 ppb ozone condition, the relative abundance of HOM dimers continually decreased from $C_{17}$ to $C_{20}$ compounds. In contrast, the $C_n$ pattern of HOM dimers in limonene SOA kept the same under different ozone concentrations, which reached a maximum for $C_{19}$ species. The different $C_n$ patterns of HOM dimers in β-pinene and limonene SOA may be due to the former ones tend to form via the combination of $C_8$ and $C_9$ or $C_9$ and $C_9$, whereas the later ones are preferred to form via the combination of $C_{10}$ and $C_9$. Such an explanation agrees with the higher averaged carbon number, oxygen number, and molecule weight of limonene SOA than β-pinene SOA (Table 1).

### 3.3 Volatility of HOMs in β-pinene and limonene SOA

HOMs in β-pinene and limonene SOA were categorized into semi-, low-, and extremely low-volatile organic compounds (SVOCs, LVOCs, and ELVOCs) for exploring their volatility distribution. Figure 7a shows that HOMs in both β-pinene and limonene SOA mainly exist as LVOCs, but their MS spectra intensity-based relative abundances ($RA_{HOMs}$) at three ozone concentrations exhibit different patterns, i.e., increased as increasing ozone concentration for β-pinene SOA, while reached the maximum value at the ozone concentration of 315 ppb for limonene SOA. Moreover, the $RA_{HOMs}$ pattern of LVOCs resembles the sum of SVOCs, LVOCs, and ELVOCs, reflecting the strong impact of low volatile species on SOA's volatility and oxidation state. Figure 7b shows that formula number-based total $RA_{HOMs}$ of SVOCs, LVOCs, and ELVOCs increased as the increasing ozone concentrations for both β-pinene and limonene SOA, with the relative abundance of LVOCs equivalent to ELVOCs. Thus, Figure 7 reflects that mass variation of β-pinene and limonene SOA during ozone chemistry are largely driven by LVOCs and ELOVCs. Moreover, the peak intensity- and formula number-based $RA_{HOMs}$ of LVOCs and ELVOCs in limonene SOA is significantly higher than β-pinene SOA, indicating a higher contribution of LVOCs and ELVOCs to the HOMs in limonene SOA. These findings agree with previous discovery of limonene ozonolysis as more efficient ELVOCs formation pathway than β-pinene (Jokinen et al., 2015). The volatility and carbon oxidation state averages of HOMs were also found to change in similar trends as that of LOVCs, ELVOCs, and ULVOCs in SOA, but the trends of β-pinene and limonene SOA are different (Figure S3).

Figure 8 shows the O/C ratio of β-pinene and limonene SOA constituents versus their estimated volatility. Figure 8a-8c shows that O/C ratio range of ULVOCs in β-pinene SOA broadened as the increase of ozone concentration. Meanwhile, both O/C ratios and relative abundance of IVOCs, SVOCs, LVOCs, and ELVOCs increased prominently, whereas the relative abundance of compounds with low O/C ratios decreased. This indicates that deeper oxidation of β-pinene may decrease the overall volatility of SOA particles via changing the relative abundance of organic matter with different volatilities. Figure 8d-8f shows that O/C ratio and volatility distribution of limonene SOA components vary slightly. This may be correlated with the preference of limonene to form highly oxygenated and low-volatile reaction products via autoxidation mechanism (Jokinen et al., 2015). Moreover, the enrichment of ELVOCs in limonene SOA confirms that limonene is more prone to form particulate

ELVOCs even ULVOCs than β-pinene via ozonolysis, which may be related to the higher reactivity of limonene due to its intrinsic two double bonds and endocyclic structure.

Figure 9 shows the molecular corridor of β-pinene and limonene SOA, which is a two-dimensional framework explaining the physicochemical properties of organic aerosols by plotting their component's volatility and molecular weight (Li et al., 2016; Xie et al., 2020b). At 50 ppb ozone concentration, substantial amounts of unique organic molecules that mainly composed of SVOCs and LVOCs were observed, the number of which accounts for 15~20% of all the identified molecules for β-pinene or limonene SOA. Smaller amounts of unique organic molecules that mainly composed of ELVOCs and ULVOCs were observed at 315 and 565 ppb ozone concentrations. This trend suggests the ozonolysis-enhanced formation of low volatile organic species for biogenic precursors. At all three different ozone concentration conditions, β-pinene SOA always contained more low volatile organic compounds (LVOCs, ELVOCs, and ULVOCs) than limonene SOA, which distributed near the sugar alcohol ($C_nH_{2n+2}O_n$) line in Figure 9 (bule dashed line), indicating a greater impact of ozonolysis on β-pinene SOA's volatility diversity. Under 565 ppb ozone condition, β-pinene SOA comprise substantial fraction of ULVOCs with molar mass of 500~800 g mol$^{-1}$. However, limonene SOA were mainly composed of SVOCs, LVOCs, and ELVOCs with molar mass of 200~600 g mol$^{-1}$. Such a compositional difference may be associated with the different partition or evolution mechanisms of HOMs and low volatility species, including the SOA aging related volatilization.

## 3.4 Potential formation mechanisms of HOM monomers and dimers in β-pinene and limonene SOA

Previous studies show that radical chemistry of organic peroxides (e.g., $RO_2 + RO_2$, $RO_2 + HO_2$, $RO_2$ isomerization, or $RO_2 +$ NO), and reaction of stabilized Criegee intermediate with carboxylic acid can produce gas-phase monomers or dimers, the low volatile fractions of which are expected to form clusters and accommodate onto particles, contributing to SOA formation (Ehn et al., 2014; Claflin et al., 2018; Shi et al., 2022). The proposed formation mechanisms of $C_{10}$ HOM monomers during the ozonolysis of β-pinene and limonene are shown in Figure 10. Therein the hydroperoxide and alkoxy chemistry might have played an important role (Tomaz et al., 2021; Shen et al., 2021; Kundu et al., 2012). There is also the probability of ozone first reacts with endocyclic double bond of limonene, which opens the chain to form alkoxy radical and then $C_{10}H_{14}O_7$. Figure 11 shows that a number of $CH_2$ homologous series of HOM monomers were found. For β-pinene SOA, $C_9H_{14}O_7$ and $C_{10}H_{16}O_7$ were found at all three ozone concentrations, whereas $C_{10}H_{14}O_7$ and $C_9H_{12}O_7$ were only found at 315 and 565 ppb ozone concentrations. For limonene SOA, $C_{10}H_{16}O_8$ and $C_{10}H_{14}O_7$ were found at all three ozone concentrations. Like previous studies (Ehn et al., 2014; Berndt et al., 2016; Brean et al., 2019), a common monoterpene oxidation product $C_{10}H_{16}O_9$ has also been observed in β-pinene SOA in this study. At high ozone concentrations, the O numbers of HOM dimers in β-pinene SOA are up to 15, while that for limonene SOA was only 13, reflecting the higher oxidation degree of HOMs in β-pinene SOA than limonene SOA-associated HOMs.

The accretion products formed from self-combination or cross-reaction of $RO_2$ radicals has been proposed to be generally important in producing higher-functionalized $RO_2$ radicals, HOM dimers, and SOA via the following pathways (Berndt et al., 2018b; Bianchi et al., 2019; Kahnt et al., 2018; Ehn et al., 2014; Berndt et al., 2018a; Tomaz et al., 2021):

$$RO_2 + RO_2 \rightarrow ROOR + O_2 \qquad (6)$$

$$RO_2 + R'O_2 \rightarrow RO_4R' \rightarrow RO\cdots O_2\cdots OR' \rightarrow ROOR' + O_2 \qquad (7)$$

$$RO_2 + R'O \rightarrow RO_3R' \qquad (8)$$

Where the $RO\cdots O_2\cdots R'O$ means a cage structure intermediate formed from the asymmetric cleavage of tetroxide and eventually converts to $ROOR'$ (Lee et al., 2016). Accretion reactions transform the mass from monomers to oligomers, yielding products with a higher number of carbon atoms and converting semi-volatile molecules to higher-molecular weight compounds that have lower saturation vapor concentrations (Barsanti et al., 2017). Accretion reactions probably exist in both gas phase and particle phase, with partial gas phase accretion products partition into the particle phase and undergo further aging process. Thus, the measured particle-phase dimers may be quite different from the original gas-phase ones (Zhang et al., 2017). The gas-phase accretion reactions have been studied under laboratory conditions and were also suggested to play an important role in ambient environment (Berndt et al., 2018a). For this study, we suggest that gas-phase reactions significantly affected the overall SOA composition especially the HOMs at different ozone concentrations. Ozone is the driving force of the SCIs formation by influencing the $^{\bullet}OH$ concentration and indirectly the formation of HOMs. High molecular weight and low volatile dimers have been identified as important components of environmental aerosols. The monomeric building blocks of the dimer esters formed through β-pinene ozonolysis are attributed to one of the dicarboxylic acids, such as $C_9H_{14}O_4$ (*cis*-pinic acid), $C_8H_{12}O_4$ (*cis*-norpinic acid), and $C_8H_{14}O_5$ (diaterpenylic acid), which can be indicators of pinene oxidation products during dimer formation (Kenseth et al., 2018). Trimer-like compounds and highly oxidized dimers are typically in the range of 450~650 Da (Kundu et al., 2012), with limonene SOA having a higher relative abundance than β-pinene SOA (Figure 4). The formation of trimers is associated with the presence of two double bonds in limonene. One of the C=C double bonds is first oxidized to the dimer products, while the other double bond provides a reaction position for further oxidation of the dimers, making it easier to form dimer $RO_2$ radicals (Guo et al., 2022). It is noticeable that the mechanism of dimer formation in similar monoterpene systems remains unresolved and warranty follow up studies (Kenseth et al., 2018).

The chemical formula together with expected molecular structures of identified dimers are shown in Table S3. Most HOMs in β-pinene and limonene SOA were very similar, while the relative abundance of different HOMs varied widely, suggesting that the reaction pathways are similar, but the degree of branching in the reaction mechanism is different. Moreover, the rate of dimer formation by self and cross-reacting $RO_2$ radicals not only depends on the structure of $RO_2$ radical, but also increases with the size of the $RO_2$ radical (Berndt et al., 2018a).

## 4 Conclusions

At lower mass concentrations, polar components seem to dominate the organic aerosols, while at higher concentrations, VOCs may condense (Grieshop et al., 2007). Environmental SOA compositions are concentration-dependent, in agreement with the

results of this study. Distribution of HOMs in SOA formed by ozonolysis is also affected by ozone concentrations. The precursors of β-pinene and limonene have the same molecular formula but different structures, with β-pinene to be found more obviously affected. The relative abundance of HOMs ($RA_{HOMs}$) in β-pinene SOA was more significantly affected by ozone concentration, while limonene SOA was less affected by ozone concentration, which was related to the high reactivity of limonene via two double bonds and endocyclic structure. β-pinene was found more prone to form HOM monomers with $C_8$ and dimers with $C_{17}$ fragments, while limonene was more inclined to form HOMs monomers with $C_{10}$ and dimers with $C_{19}$ subgroups, which is related to the loss of a formaldehyde and subsequent oxidation of β-pinene by the addition of $O_3$ to the exocyclic double bond. In addition, distinct volatilities and abundances of HOMs in β-pinene and limonene SOA reflects the different molecular response of particulate reaction products to biogenic precursor oxidation, leading to different SOA size and number distribution profiles. Higher ozone concentrations (315 and 565 ppb) were favorable for the formation of HOMs and ELVOCs, and the number of unique organic molecules was higher. Compared to β-pinene SOA, the abundance of ELVOCs in limonene SOA is higher.

In addition to ozone, other oxidants in the atmosphere may also impact the formation and evolution of HOMs. For example, $NO_2$ inhibited the formation of highly oxidized dimer products by suppression of autoxidation (Rissanen, 2018). When considerable amount of ˙OH was present, the potential dimer sources can also be suppressed (Zhang et al., 2017). Moreover, Simon et al. found a continuously decreased oxidation level of α-pinene SOA and yields of HOMs as the temperature decreased from 25 to -50 °C (Simon et al., 2020). Similar result was also confirmed in urban field samples (Brean et al., 2020). Whether $O_3$ can exhibit a synergistic effect with other different oxidants and environmental conditions (e.g., humidity and temperature) in influencing HOMs formation and evolution is worthy to be explored in follow up studies. Current mass spectrometry techniques can only obtain information on the molecular formula of the HOMs, hindering the study of the detailed formation mechanism of many atmospheric precursors. Therefore, experimental conditions and new analytical techniques need to be developed to characterize HOMs rapidly and in detail (Bianchi et al., 2019).

*Data availability.* The dataset for this paper is available upon request from the corresponding author (fupingqing@tju.edu.cn).

*Author contribution.* DL participated in the investigation, methodology, software development, formal analysis, and writing of the original draft. SZ and SC participated in the methodology and formal analysis. YZ collected the samples. All co-authors participated in validation as well as in reviewing and editing of the manuscript. PF and HT participated in the conceptualization, project administration, and funding acquisition.

*Competing interests.* The authors declare that they have no conflict of interests.

*Acknowledgments.* This work was supported by the National Natural Science Foundation of China (NSFC) (Grant Nos. 42130513 and 41625014), Tianjin Research Innovation Project for Postgraduate Students (Grant No. 2021YJSB135), Max Planck Institute for Chemistry, and Helmholtz-Zentrum Hereon. The authors thank Ulrich Pöschl for stimulating discussions.

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

**Table 1.** The chemical characteristics of β-pinene SOA and limonene SOA samples.

| [O$_3$] / ppbs | MW | O | O/C | DBE | OS$_C$ | Volatility fractions (%) | | | | |
|---|---|---|---|---|---|---|---|---|---|---|
| | | | | | | ULVOCs | ELVOCs | LVOCs | SVOCs | IVOCs |
| β-Pinene SOA | | | | | | | | | | |
| 50 | 305 | 4.77 | 0.30 | 4.69 | -0.98 | 14 | 28 | 33 | 17 | 8 |
| 315 | 307 | 5.49 | 0.35 | 4.40 | -0.89 | 24 | 23 | 28 | 17 | 8 |
| 565 | 319 | 5.93 | 0.37 | 4.54 | -0.84 | 30 | 23 | 23 | 16 | 8 |
| Limonene SOA | | | | | | | | | | |
| 50 | 326 | 6.07 | 0.36 | 4.78 | -0.82 | 12 | 21 | 30 | 25 | 12 |
| 315 | 349 | 6.88 | 0.39 | 4.99 | -0.76 | 26 | 21 | 22 | 21 | 10 |
| 565 | 339 | 6.58 | 0.38 | 4.87 | -0.78 | 26 | 21 | 23 | 20 | 10 |

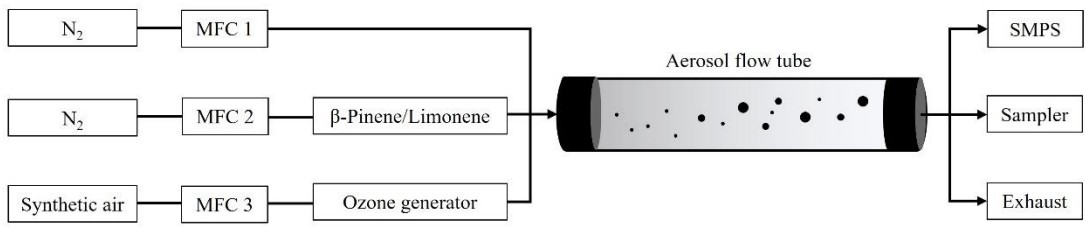

**Figure 1.** Schematic of the experimental setup for generation and collection of SOA. MFC: mass flow controller. SMPS: scanning mobility particle sizer.

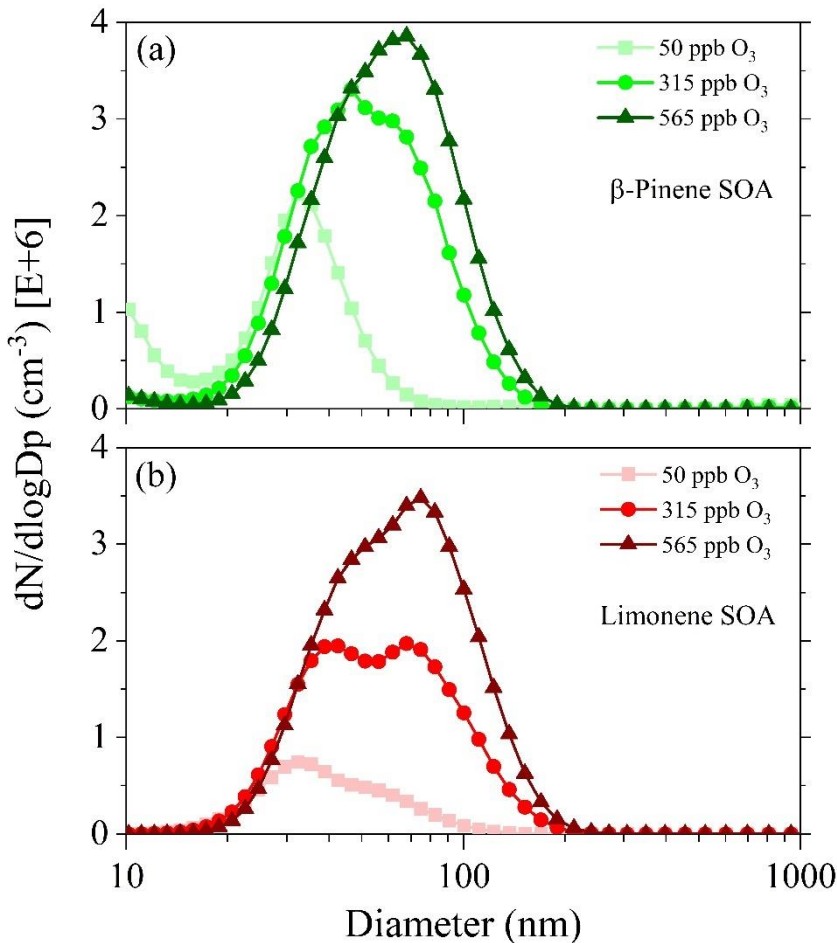

**Figure 2.** Particle size and number concentration distributions of β-pinene SOA (a) and limonene SOA (b) from SMPS measurement.

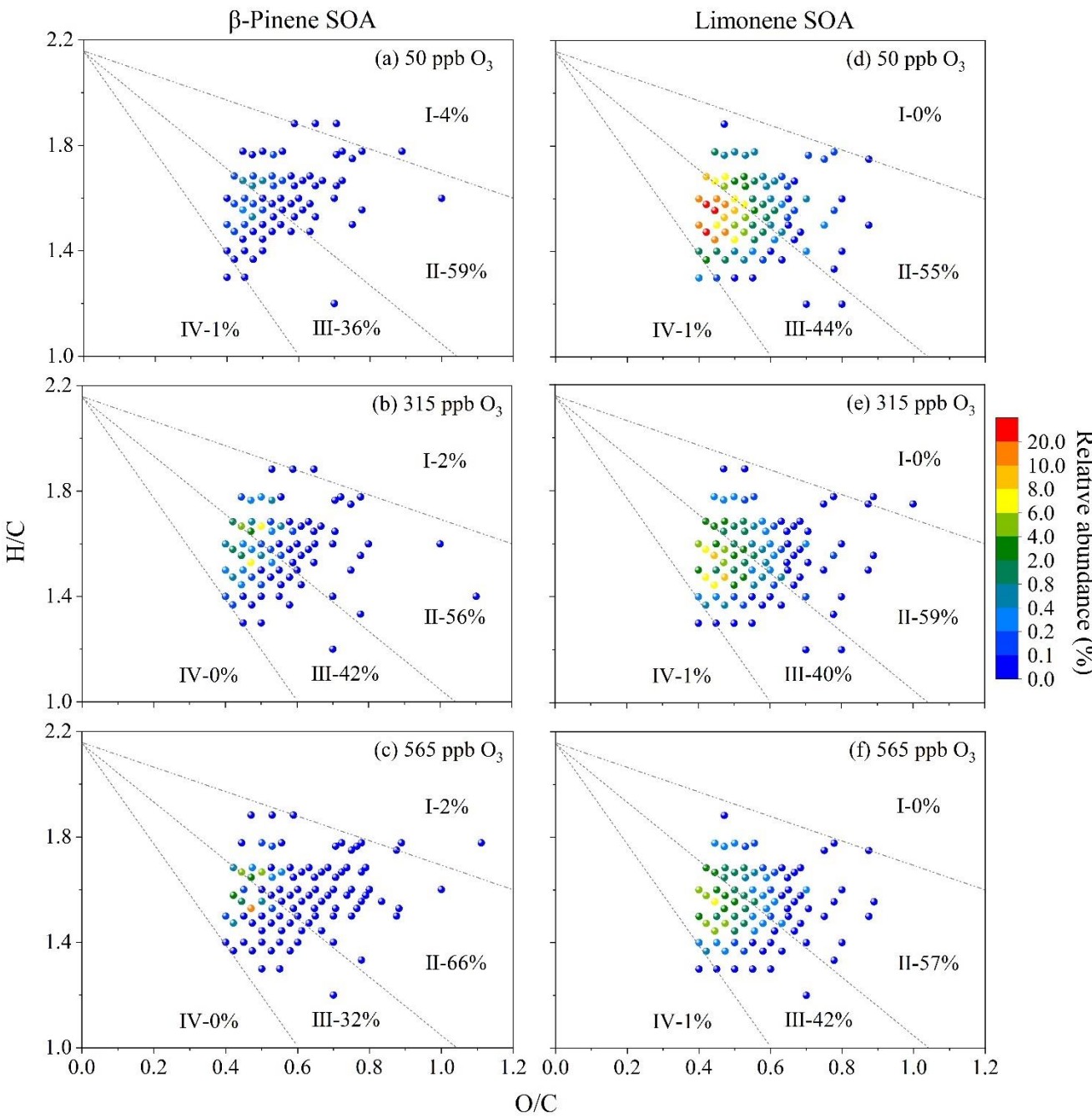

**Figure 3.** MCR-VK diagrams of HOMs in β-pinene SOA (a, b, c) and limonene SOA (d, e, f) that formed at different ozone concentrations. The colors of the dots indicate the relative abundance of compounds.

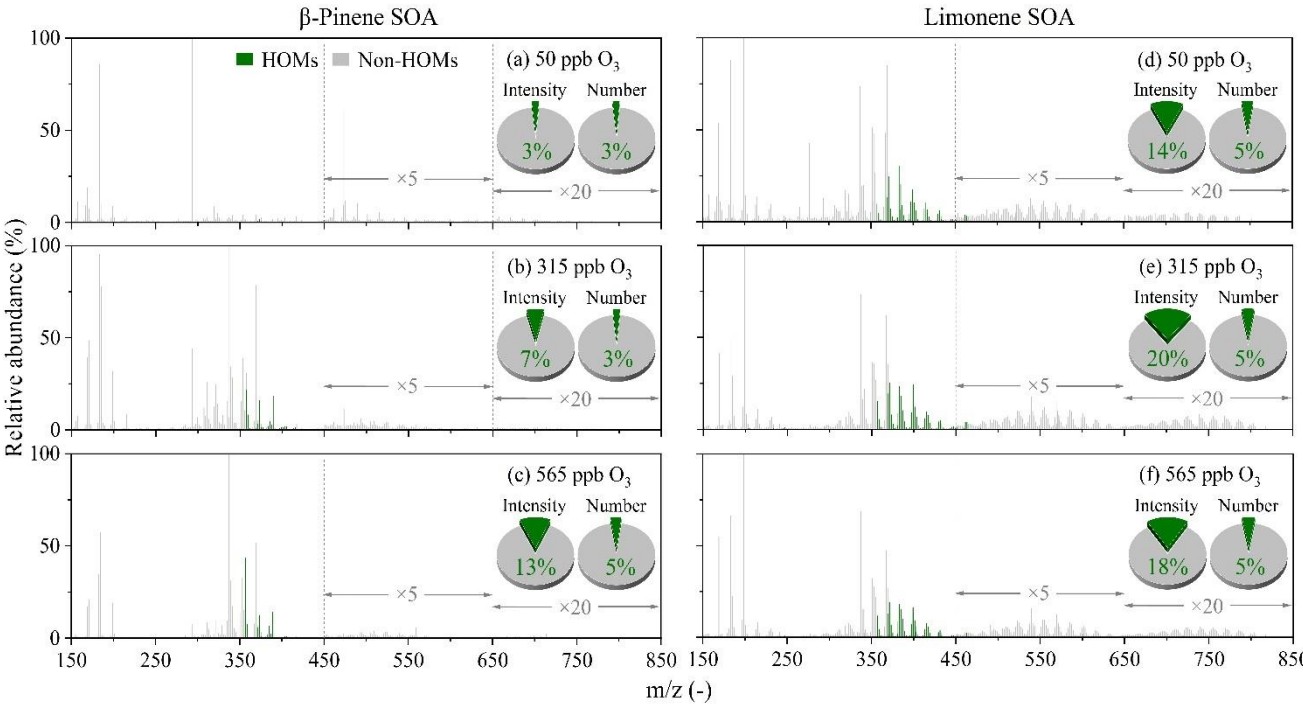

**Figure 4.** Mass spectral fingerprint and relative abundances of HOMs (green) and non-HOMs (gray) in β-pinene SOA (a, b, c) and limonene SOA (d, e, f) that formed at different concentrations of ozone. The relative abundances of compounds with m/z 450~650 and 650~850 were increased by factors of 5 and 20, respectively. The pie charts indicate the ion intensity- and ion number- based relative abundance of different compounds.

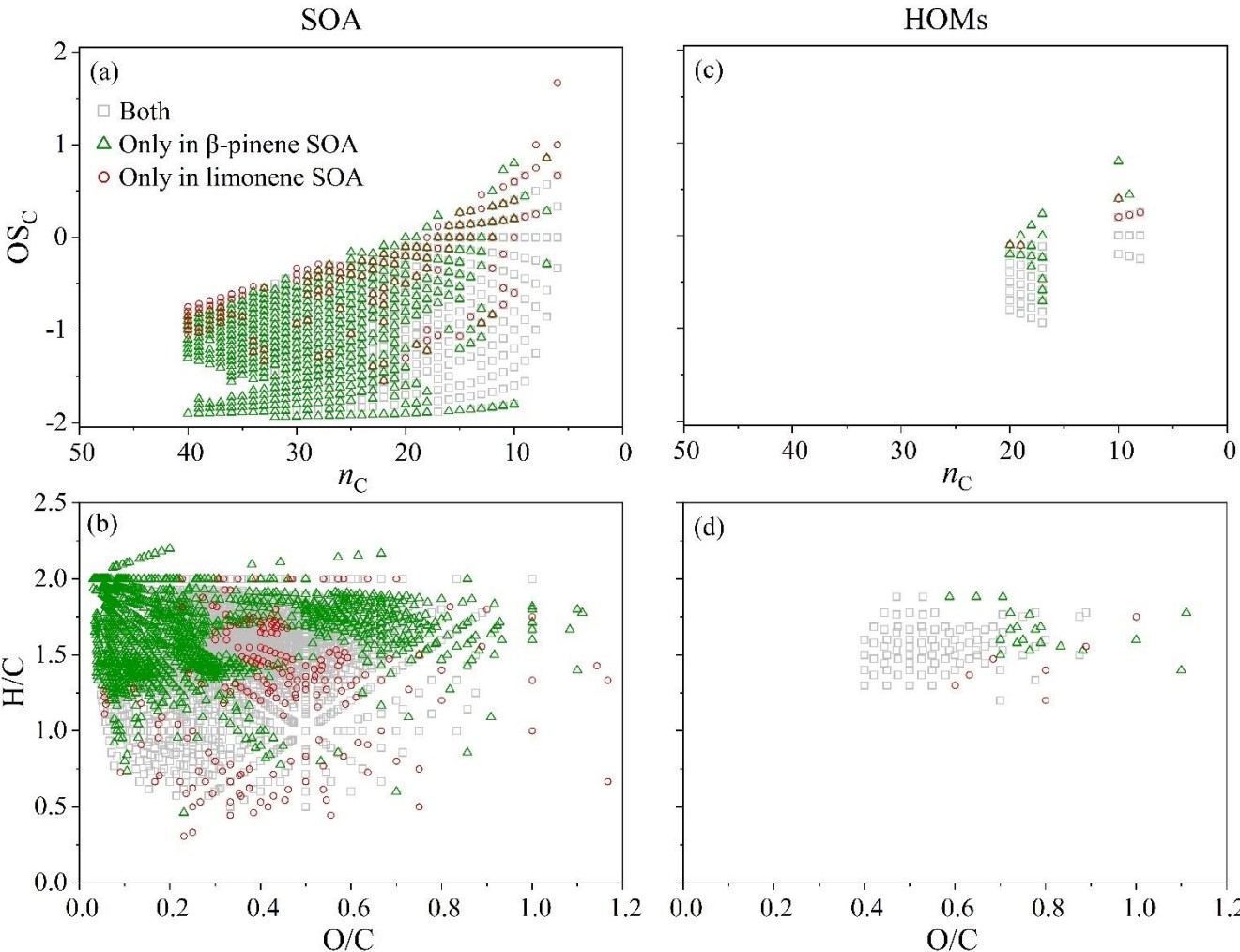

**Figure 5.** The carbon oxidation state ($OS_C$) vs. the number of C atoms ($n_C$) (a, c) and Van Krevelen diagrams (b, d) for the unique assigned formulas from β-pinene SOA and limonene SOA and the corresponding HOMs (c, d). Gray squares represent assigned formulas observed both in β-pinene SOA and limonene SOA. Green triangles and red circles represent assigned formulas observed only in β-pinene SOA and limonene SOA, respectively.

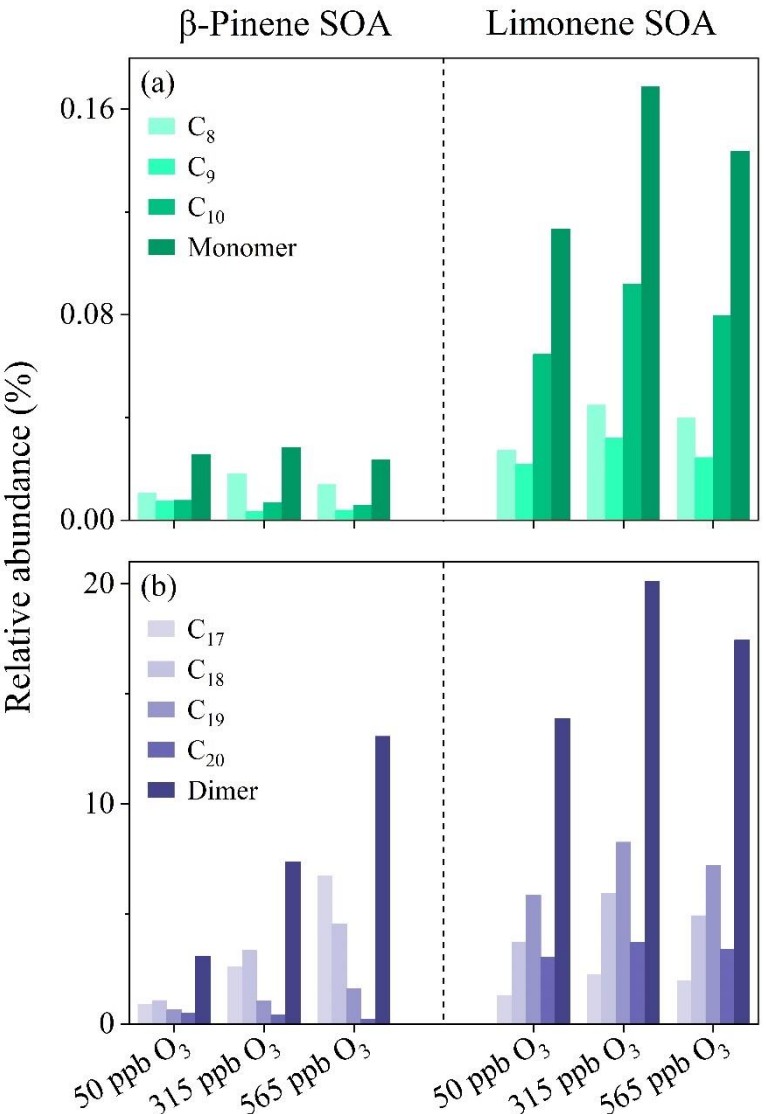

**Figure 6.** The relative abundance of HOMs monomers (a) and dimers (b) identified in β-pinene and limonene SOA as a function of carbon number ($C_n$).

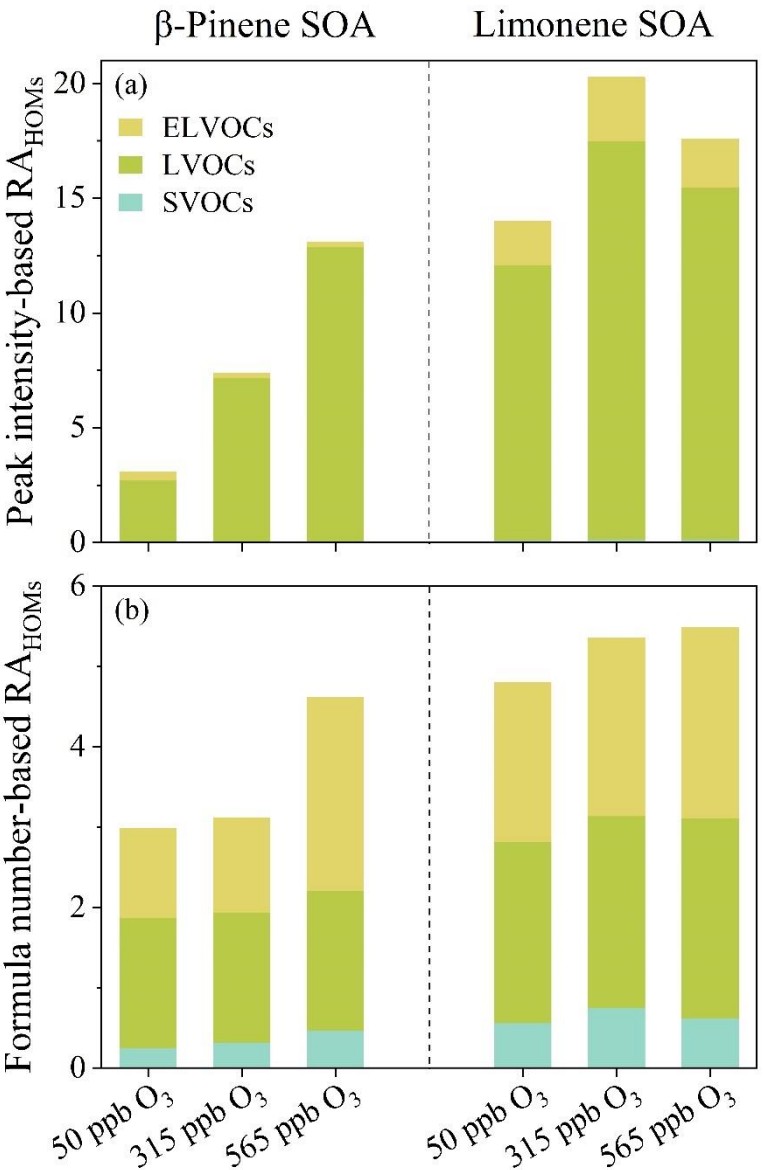

**Figure 7.** The peak intensity- (a) and formula number- (b) based relative abundance of HOMs with different volatilities in SOA particles produced from ozonolysis of β-pinene at and limonene under the three ozone concentrations.

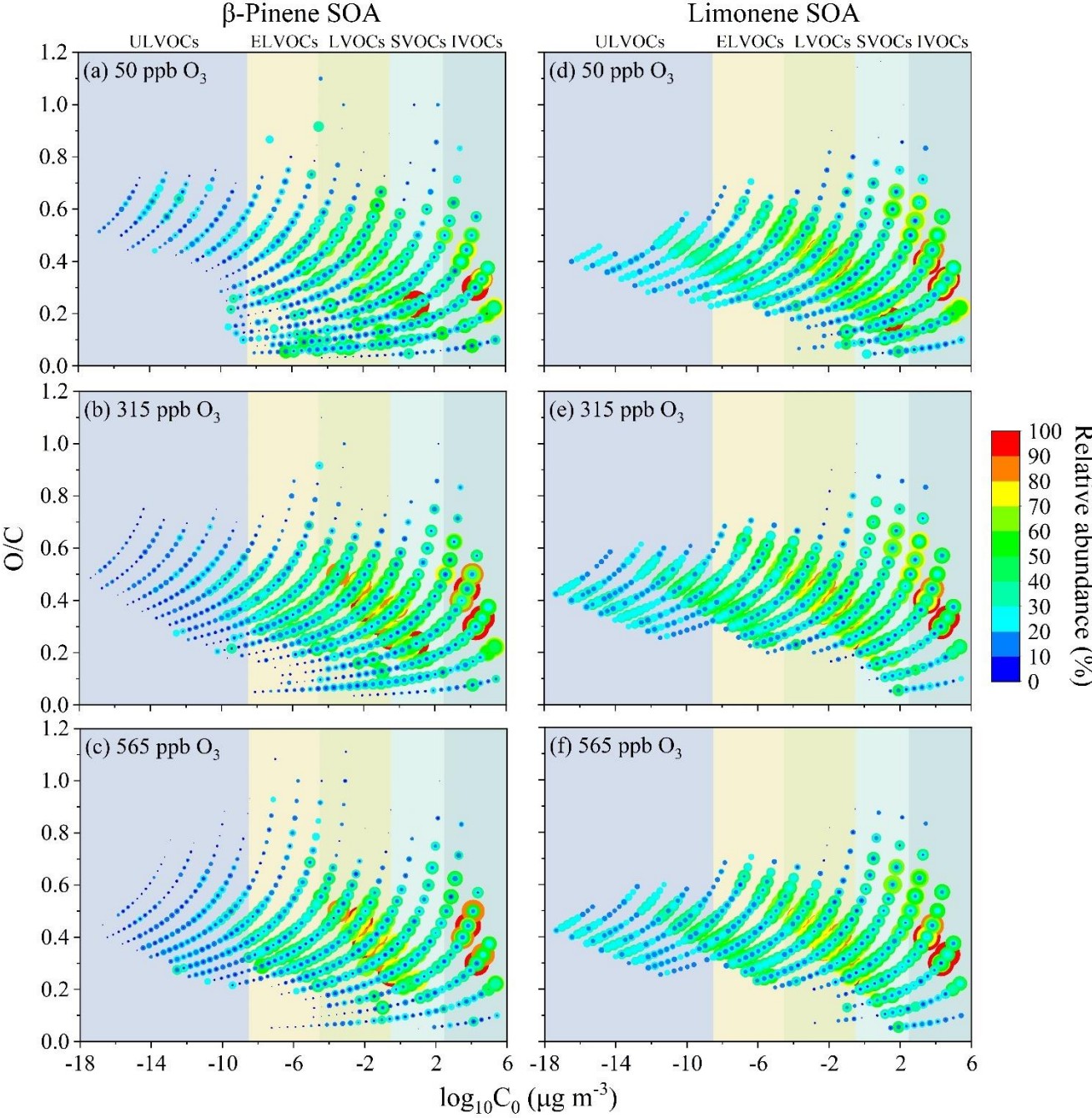

**Figure 8.** The pure compound saturation mass concentration ($C_0$) and the atomic oxygen and carbon ratios (O/C) of β-pinene SOA (a, b, c) and limonene SOA (d, e, f). The color-codes and the size of the dots represent the relative abundance on a logarithmic scale.

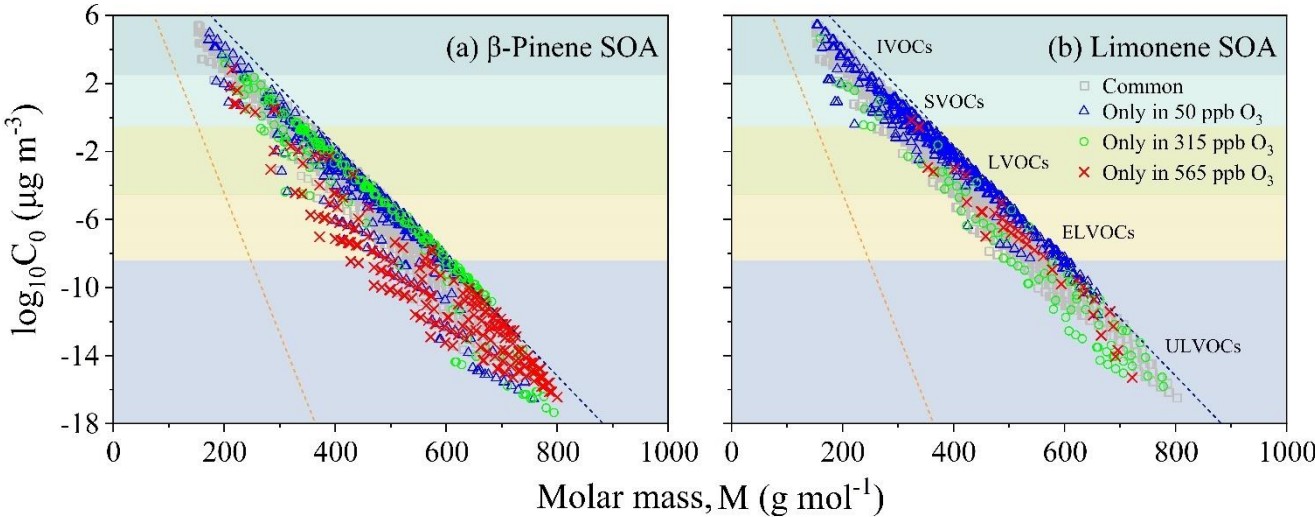

**Figure 9.** Molecular corridors of β-pinene (a) and limonene SOA (b) formed at 50 ppb, 315 ppb and 565 ppb ozone conditions. The gray squares represent the common compounds identified in SOA formed at three different ozone concentrations. The blue triangles, green circles, and red crosses indicate compounds only found in 50, 315, and 565 ppb ozone, respectively. The dotted lines represent linear $n$-alkanes $C_nH_{2n+2}$ (blue with O/C = 0) and sugar alcohols $C_nH_{2n+2}O_n$ (orange with O/C = 1).

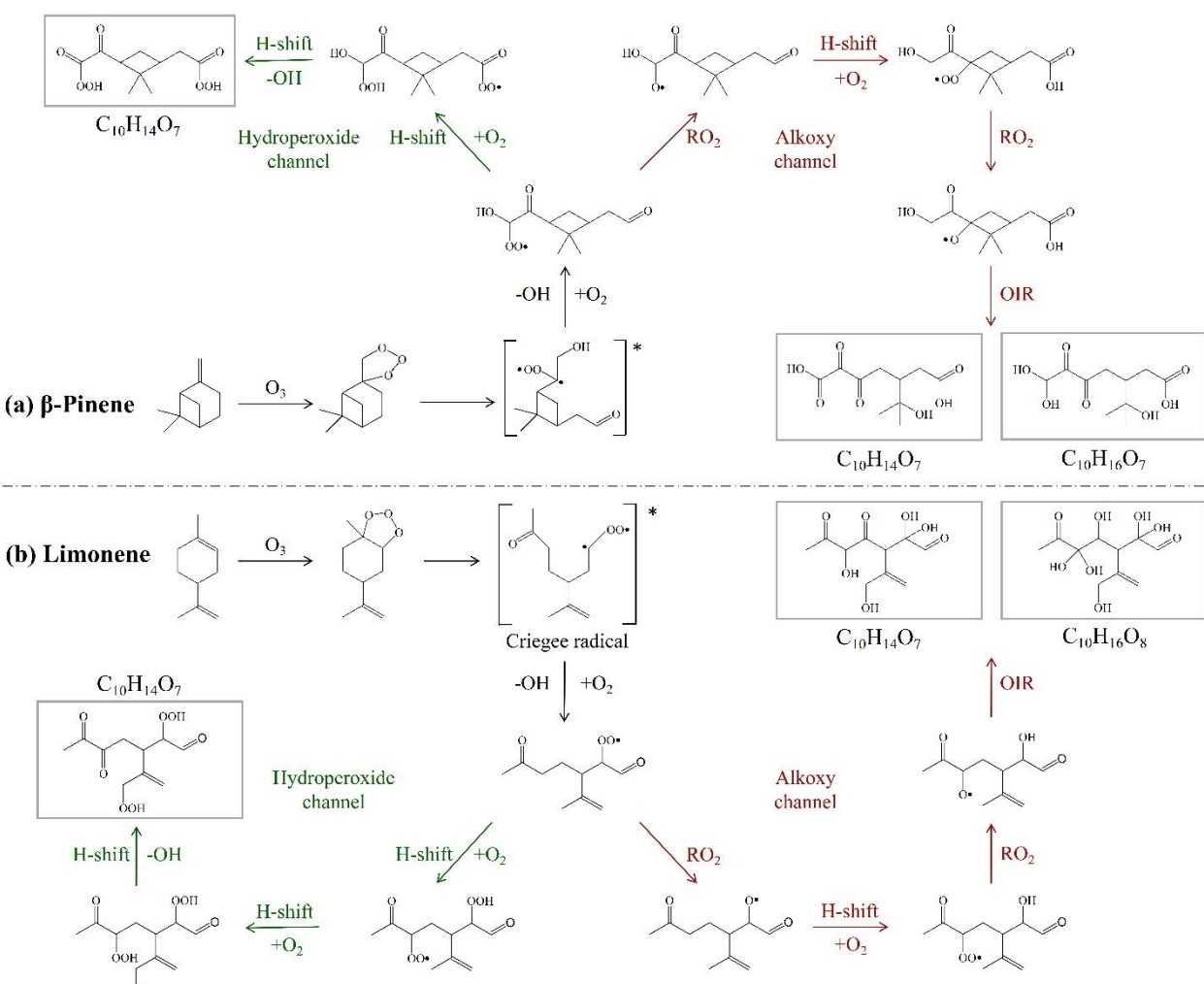

**Figure 10.** Proposed formation mechanism of $C_{10}$ HOMs from the β-pinene and limonene ozonolysis via hydroperoxide channel (green) and oxygen-increasing reactions (OIR) (H-shift $\rightarrow O_2 \rightarrow RO_2$) of alkoxy channel (red) (Tomaz et al., 2021; Shen et al., 2021; Kundu et al., 2012). The $C_{10}H_{14}O_7$ organic molecule in the gray rectangle are the hypothesized structures.

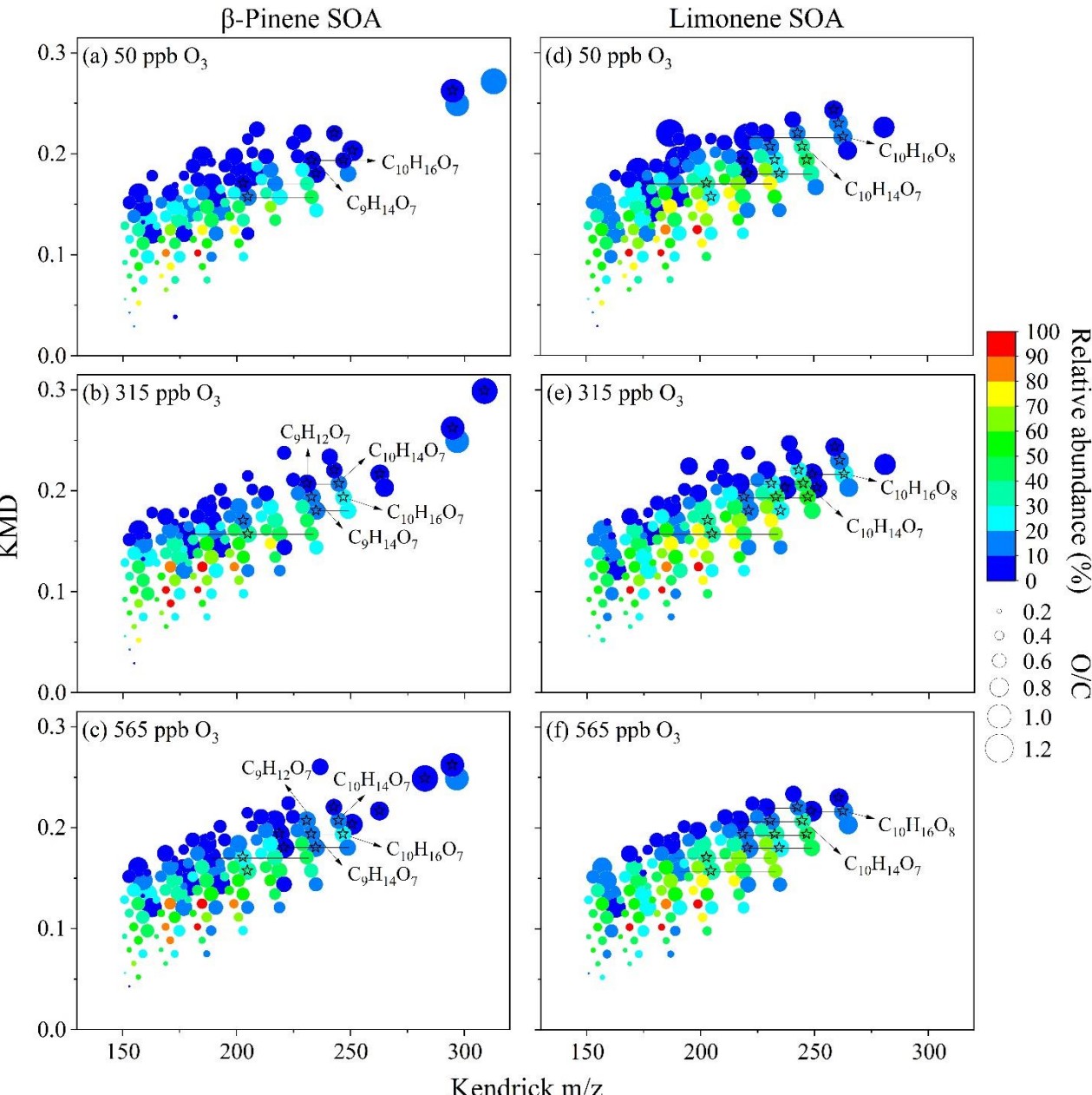

**Figure 11.** Kendrick mass defect of detected organic compounds in β-pinene SOA (a, b, c) and limonene SOA (d, e, f). Different colors represent the logarithm of relative abundance. Different dot sizes denote the atomic ratio of oxygen to carbon (O/C). The black pentagrams delegate HOMs.