# Peer review of "Large differences of highly oxygenated organic molecules (HOMs) and low volatile species in SOA formed from ozonolysis of $\beta$ -pinene and limonene"

_Atmospheric Chemistry and Physics, 2022_

## Author Comment (AC1)

**Response to the comments of Anonymous Referee #1**

D. Liu and co-workers have used filter sampling and mass spectrometry to study the composition of secondary organic aerosol (SOA) material formed from the ozonolysis of beta-pinene and limonene at different ozone concentrations. The topic is in principle highly atmospherically relevant, and the manuscript contains many interesting and publishable results. Unfortunately, the manuscript is in several places quite poorly written, and it is in places difficult for the reader to extract the genuinely novel results from the presented data. I'm happy to recommend publication of this manuscript in ACP, but only after a rather thorough revision including substantial copy-editing and scientific proofreading.

The overall problem in reading the manuscript is that a substantial fraction of its sentences sound sensible, exact and scientific at a quick glance, but on closer inspection are at best very vague, and at worst ill-defined, illogical or meaningless. Sometimes the actual intent and meaning of the writers can be deduced by clever guesswork (e.g. by correcting some rather trivial word order issues), but in other cases even a thorough and repeated reading leaves the reader confused. The following examples are not intended to be an exhaustive list, but simply illustrative examples of this overall problem (note: I mostly gave up on listing problematic sentences after the first 3 pages or so, so the examples are weighted toward the beginning of the text - while the introduction is arguably the most problematic part of the paper this should not be construed to mean that the results - section is problem-free):

Response: We thank the reviewer for providing detailed and constructive comments. Our point-by-point responses are shown in blue color as below.

Line 17, "abundance", please be more specific - exactly what abundance is meant

Response: The "abundance" means 'the ratio of summed mass spectrometry peak intensity of HOMs to total organic compounds that with assigned formula', which is clarified in L17-18.

-Line 18, "prefer to stabilise". As written, this would imply that the HOM molecules themselves would preferentially be stabilised (e.g. collisional, thermally, kinetically etc) at some certain [$O_3$], but what is meant is that their yield saturates with respect to [O3] -not the same thing.

Response: we changed the "stabilise" to 'saturate' in L20 and L21.

-Line 23, "formation of compounds with 10 carbon atoms". As the precursors themselves contain 10 carbon atoms, no mechanism is needed for their formation. Presumably the authors mean the formation of compounds with 10 C atoms AND SOME NUMBER OF OATOMS.

Response: to avoid confusion, we have changed this phrase to 'formation of oxygenated organic compounds with 10 carbon atoms' (L24).

-Line 38: "wide range of volatility, which has a strong temperature-dependence". What is meant by this - that an individual saturation vapor pressure has a strong temperature dependence (generally true), that the variation/range in volatility has a temperature dependece, or perhaps both? Whatever is meant, what is the relevance of the temperature-dependence to the rest of the discussion?

Response: You are right, we removed the 'which has a strong temperature-dependence' to avoid redundancy and irrelevance.

-Line 39: "ambient species are thought to consist mainly of low volatility species", apart from the grammar issue (species…species) this is not even true as written - the ambient air certainly contains a large range of species (as indeed argued in the previous sentence!) all the way from fully volatile to ULVOC. I assume the authors do have an actual argument here, the presentation is just missing some key assumptions or concepts (e.g., do they mean the species present in ambient SOA?).

Response: We revised this sentence as follows: 'biogenic SOA comprise thousands of organic compounds, which exhibit a wide range of volatilities (Donahue et al., 2012; Ehn et al., 2014; Simon et al., 2020)' (L60-61).

-Line 43 "ozonolysis as the most effective", "as" should presumably be "is", and also this sentence needs some caveats - for which compounds and in which conditions is ozonolysis the most effective, and compared to what (presumably OH and NO3 oxidation)?

Response: Agree. We have revised these sentences as follows: 'Gas phase ozonolysis, hydroxyl radical chemistry, and nitrate radical ($NO_3$) oxidation etc. have been found as effective formation pathways for biogenic SOA (Kroll and Seinfeld, 2008; Kirkby et al., 2016), with the partition of low volatile organic compounds on exiting seed particles or homogeneous nucleation as key particle formation pathway (Donahue et al., 2012; Saukko et al., 2012)' (L49-52).

-Line 45, "BVOCs forming ULVOC even in the absence of $H_2SO_4$", it's not the ULVOC formation in the absence of $H_2SO_4$ that is the surprising fact, but the production of aerosol particles in the absence of $H_2SO_4$. (Here I again assume the authors do actually mean the latter, they just wrote the sentence incorrectly.)

Response: You are right. We revised the sentence as follows: 'Recent studies also showed that oxidation of BVOCs can produce large amounts of SOA particles via the nucleation of ULVOCs with the absence of sulfuric acid (Kirkby et al., 2016; Guo et al., 2022)' (L67-69).

-Line 52, "high oxygen-containing but low oxidation state determining the oxidation potential", here I have no idea what the authors mean, this seems to make no sense.

Response: We removed this sentence to avoid confusion.

-Line 54, "HOMs refer to the process" - no, HOMs refer to a certain subgroup of the products of this process (and indeed that seems to be the definition the authors are using)

Response: We agree and have revised this sentence as follows: 'For instance, peroxyl radicals ($RO_2$) can undergo an intramolecular hydrogen atom shift (H-shift) to form a hydroperoxide functionality (HOO-) and an alkyl radical (RO), and then molecular oxygen rapidly attaches to form a new more oxidized $RO_2$ radical, and be repeated several times to form HOMs (Bianchi et al., 2019)' (L89-92).

-Line 57, "precursors decomposing into aerosols", certainly this is not what is going on

Response: We revised it as follows: 'Iyer et al. suggested that after a single oxidant attack, BVOCs can be oxidized to low-volatility species on sub-second timescales, which consequently undergo decomposition or bear new particle formation (Iyer et al., 2021)' (L66-67).

-Line 58, "Laboratory studies of HOMs observed by ozonolysis of monoterepenes are closely corresponded to"… this phrasing makes no sense.

Response: We removed this sentence to avoid confusion.

-Line 60-65: the science is correct here, the phrasing is just very very poor.

Response: Thanks. We discussed the HOMs in the next paragraph. We improved the writing on definition of VOCs, IVOCs, SVOCs, LVOCs, ELVOCs, and ULVOCs as follows: 'Based on grouping the estimated effective saturation mass concentrations $C_0$ (Schervish and Donahue, 2020), the volatility of organic aerosols has been categorized into volatile organic compounds (VOCs, $C_0 > 3\times10^6$ µg m$^{-3}$), intermediate volatile OC (IVOCs, $300 < C_0 < 3\times10^6$ µg m$^{-3}$), semivolatile OC (SVOCs, $0.3 < C_0 < 300$ µg m$^{-3}$), low-volatile OC (LVOCs, $3\times10^{-5} < C_0 < 0.3$ µg m$^{-3}$), extremely low-volatile OC (ELVOCs, $3\times10^{-9} < C_0 < 3\times10^{-5}$ µg m$^{-3}$), and ultralow-volatile organic compounds (ULVOCs, $C_0 < 3\times10^{-9}$ µg m$^{-3}$), respectively (Hallquist et al., 2009; Simon et al., 2020)' (L61-66).

-Line 66, "HOMs can provide nucleation conditions for early growth": no, the HOMs can nucleate (or not), or participate in early growth (or not), this phrasing about providing conditions is meaningless

Response: Agree. We have changed it to 'playing an important role in the early growth of atmospheric organic aerosols (Ehn et al., 2014; Wang et al., 2020)' (L77).

-Line 70 "an exocyclic bond as the second-most-abundant VOC" - no it's beta-pinene (not its exocylic double bond) that's the VOC

Response: To avoid confusion, we have changed it to 'β-pinene and limonene are typical and important biogenic precursors that are released approximately 30.3 Tg yr$^{-1}$ globally (Guenther et al., 2012). The molecular composition of these two compounds contains the same number of carbon atoms, hydrogen atoms, and double bond equivalents (DBE). However, β-pinene has a bicyclic structure with an exocyclic double bond, and limonene has a monocyclic structure with an exocyclic double bond and an endocyclic double bond, which is more reactive than the frond one (Gallimore

et al., 2017; Kenseth et al., 2018)' (L38-42).

-Line 80 "nucleation rate of monoterpene SOA" - SOA is by definition material that has already formed particles (e.g. nucleated), what is presumably meant (though not measured here directly) is the nucleation rate of monoterpene-derived SOA precursors, such as the HOM discussed above. These are not the same thing.

Response: This sentence is changed to: 'The yield of monoterpene SOA also strongly dependents on oxidant types and concentrations' (L53-54).

-Line 94, "obtain reaction mechanism", this is incorrect - the mass spectra just gives the molecular compositions, these can then often be used to indirectly infer something about the reaction mechanism but claiming that the mechanism is "obtained" is wrong.

Response: Thanks. The sentence is changed to 'Then Fourier transform ion cyclotron resonance mass spectrometer equipped with a 7 Tesla superconducting magnet (7T FT-ICR MS) was used to study the molecular composition and formation mechanism of SOA' (L105-106).

-Line 207: "high ozone concentration tends to increase oxygen reaction", this seems to be almost trivially true - maybe the authors need to specify what is meant by "oxygen reaction" here.

Response: We corrected this sentence by changing it to 'High concentration of ozone tends to convert less oxidized organic molecules to highly oxygenated organic molecules (HOMs) via oxygen-increasing-reactions on the carbon skeleton, i.e., addition of oxygen atom to the intermediate alkoxy radicals to form new alkoxy radicals (Kundu et al., 2012)' (L236-239).

-Line 208, "may indicate the importance of ozone… by ozonation". Ozone is by definition important in ozonation, so this sentence as written is tautologically true (and thus meaningless).

Response: We changed this sentence 'This reflects the importance of ozone concentration in determining redox activity of SOA' (L241-242).

-Line 215, "peak intensity of MW" - presumably what is meant is the MW (molecular weight) corresponding to the peak of highest intensity (not the same thing).

Response: We deleted this term.

-Line 215, "maximum proportion", what is meant by proportion here?

Response: We deleted this term and improved the discussions in L250-256.

-Line 235, "limonene is preferred to proceed oxygenate and accretion reaction", what is meant by this?

Response: This sentence has been moved and integrated in the current L242-249: 'For limonene SOA, when the ozone concentration was increased from 315 to 565 ppb, the MW, O atom and DBE are decreased, indicating that high carbon- and oxygen-containing organic molecules in limonene SOA may fragment at high ozone concentration to form low carbon number and less oxidized organic molecules. The element characteristic values of limonene SOA were generally higher than β-pinene SOA, probably due to the following reasons. First, ozonolysis of limonene proceeds in a faster rate ($k_{\text{limonene+O}_3} = 2.1 \times 10^{-16}$ cm$^3$ molecules$^{-1}$ s$^{-1}$) than β-pinene ($k_{\beta\text{-pinene+O}_3} = 1.5 \times 10^{-17}$ cm$^3$ molecules$^{-1}$ s$^{-1}$) (Atkinson and Arey, 2003). Second, limonene is more inclined to undergo oxygenate and accretion reactions than β-pinene. Third, non-condensation reactions might play a more important role in the limonene SOA formation (Kundu et al., 2012)'.

-Line 279, "front" should presumably be "former"

Response: We changed "front" to 'former' for the whole manuscript.

-Line 323-326, The first sentence here lists possible gas-phase formation pathways for (HOM) monomers and dimers. The second sentence then mentions rapid particle-phase reactions of some of the monomer and dimer types. The third sentence says "Due to the high activity of these pathways, the dimers…are expected to be distributed directly onto particles after gas-phase production". While I agree that the dimers will be "distributed directly onto particles", surely this is due to their low volatility, and not the rapidity of the subsequent reactions? How could a dimer still in the gas phase be affected by the reactions going on in the particle phase?

Response: To avoid confusion, we removed the discussion on particle phase chemistry of dimers and improved the first sentence as follows: 'Previous studies show that radical chemistry of organic peroxides (e.g., $RO_2 + RO_2$, $RO_2 + HO_2$, $RO_2$ isomerization, or $RO_2 + NO$), reaction of $RO_2$ with monoterpenes, and reaction of stabilized Criegee intermediate with carboxylic acid can produce gas-phase monomers or dimers, the low volatile fractions of which are expected to form clusters and accommodate onto particles, contributing to SOA formation (Ehn et al., 2014; Claflin et al., 2018; Shi et al., 2022)' (L359-362).

-Line 339, "monomers with CH2", what is meant by this?

Response: We replaced it with the term of 'CH$_2$ homologous series' (L366).

-Line 347, "The presence of…" this whole sentence does not seem to make sense – please rephrase, rewrite and give more background as this is potentially a quite important point! The next sentence is also very long and hard to parse - please split it up into two or more sentences.

Response: We improved this paragraph as follows: 'The accretion reaction $RO_2 + R'O_2 \rightarrow ROOR' + O_2$ formed by self-combination or cross-reaction of $RO_2$ radicals has been proposed to be generally effective (Berndt et al., 2018b; Bianchi et al., 2019; Kahnt et al., 2018; Ehn et al., 2014; Berndt et al., 2018a). Tomaz et al. found that the reaction between $C_{10}H_{15}O_6$ and $C_{10}H_{15}O_8$ radicals may also contribute to the formation of $C_{20}H_{30}O_{12}$ dimer (Tomaz et al., 2021). Alternatively, a weakly bound

RO⋯O$_2$⋯R'O cage formed by the asymmetric cleavage of the tetroxide, which then loses O$_2$, intersystem crosses and alkoxy recombines, also converts to ROOR' (Lee et al., 2016). The combination of an acylperoxy radical (RC(O)O$_2$) associated with cis-pinic and a RO$_2$ related to 7- or 5-hydroxypinonic acid allows for a RO$_2$ + R'O → RO$_3$R' radical termination reaction, which is also an important pathway for the formation of HOM dimers (Kahnt et al., 2018). C$_{17}$H$_{26}$O$_8$ may be produced by the decomposition of C$_{19}$H$_{28}$O$_{11}$ with a loss of ketene from the internally contained labile trioxide function group, and the conversion of the unstable acyl hydroperoxide groups to carboxyl groups (Kahnt et al., 2018). According to recent findings, the main pathway for the formation of accretion product ROOR' is the RO$_2$ + α-pinene reaction, rather than the RO$_2$ + R'O$_2$ reaction (Shi et al., 2022). The alkyl radicals produced by the former reaction can also produce HOM dimers and trimers through an autoxidation chain. Even if the RO$_2$ radical is a RC(O)O$_2$, it is still prone to react with α-pinene. The chemical formula together with expected molecular structures of identified dimers are shown in Tables S3. Most HOMs in β-pinene and limonene SOA were very similar, while the relative abundance of different HOMs varied widely, suggesting that the reaction pathways are similar, but the degree of branching in the reaction mechanism is different. Moreover, the rate of dimer formation by self and cross-reacting RO$_2$ radicals not only depends on the structure of RO$_2$ radical, but also increases with the size of the RO$_2$ radical (Berndt et al., 2018a)' (L373-389).

-Line 375, "almost hardly", just "hardly" is enough

Response: We have changed it to 'less'. (L413)

-Line 378, "related to the way of broken bonds" please rephrase

Response: We have changed it to 'related to the loss of a formaldehyde and subsequent oxidation of β-pinene by the addition of O$_3$ to the exocyclic double bond'. (L416-417)

**Other technical or notation-related questions and suggestions:**
-Please note in the abstract that measurements were made from filter samples (as many readers may initially assume the gas phase is being probed).

Response: We added the phrase 'in aerosol filter samples' in L16.

-Line 17 in the abstract, "5-13% higher than" - should there be a comma here? Without the comma this literally means that the abundance (see above for a note on this term) of HOMs in limonene SOA was 5 to 13 percentage points higher than the abundance of HOMs in beta-pinene SOA (a claim not actually validated by the numbers in the text). With a comma this just more vaguely implies that the former is in general higher than the latter (which would seem to be true).

Response: Thank you. We changed the range to '14~20%' and added a comma after it (L18).

-Line 90, what is meant by "gradient concentrations"? Just that there were three different concentrations? OR something else?

Response: We changed the "gradient" to 'different' (L104).

-Do the authors expect ROOR or ROOH - type compounds to fragment in their ionisation setup, as recently predicted even for milder chemical ionisation (https://doi.org/10.5194/amt-15-1811-2022)? If not, why not?

Response: Nam et al. found that up on interaction with iron ions, the O-O bond of hydroperoxides containing electron-donating tert-alkyl groups such as tert-butyl hydroperoxide and 2-methyl-1-phenyl-2-propyl hydroperoxide tends to be cleaved homolytically, whereas electron-withdrawing substituents such as an acyl group in m-chloroperoxybenzoic acid facilitates O-O bond heterolysis (Nam et al., 2000). Our previous study also found that ROOH rather than ROOR is prone to decompose in water to form aqueous radicals (Tong et al., 2016). Thus, we expect that ROOH-type compounds are easier to fragment than ROOR-type compounds during their ionization process.

-Line 145, please redefine BDE here (it's defined in the intro but that is easily missed)

Response: We have added 'double bond equivalents (DBE)' in L40 and L163-164.

-What are the Int_i weights used in equation 4? I couldn't find the actual numbers anywhere. Also on line 205 the weighed average is mentioned with reference to Table 1, but I don't see the weighted average numbers in that table.

Response: Int_i is the mass spectra peak intensity of each individual formula for corresponding samples (L189). We removed the 'weighted' to avoid confusion.

-What are the likely error margins of the (useful but crude) volatility estimate of equation 5? At least a couple of orders of magnitude I assume, since functional group identities are completely ignored?

Response: Using $C_{19}H_{28}O_8$ compound as an example, for each additional hydroperoxide groups(-OOH) there is a difference in volatility of 0.6~0.8 orders of magnitude. However, the effect of functional groups on volatility values is not considered in this paper.

| Compound | $Log_{10}C_0$ ($\mu g\ m^{-3}$) |
|---|---|
| $C_{19}H_{28}O_8$ | -2.83699 |
| $C_{19}H_{27}O_6(OOH)$ | -1.98218 |
| $C_{19}H_{26}O_4(OOH)_2$ | -1.19147 |
| $C_{19}H_{25}O_2(OOH)_3$ | -0.47287 |
| $C_{19}H_{24}(OOH)_4$ | 0.16422 |

-Line 204, "plausible different partition and agglomeration kinetics", can you be a little more specific, and e.g. suggest which mechanisms could lead to the observed difference?

Response: We improved this sentence as follows: 'The precursor-dependent number- and mass-size distribution profiles in Figures 1 and S1 may be related to different partition and agglomeration

kinetics of low volatile organics, with the former process playing a plausible stronger role. The "partition" here means an equilibrium between the absorption and desorption rates of oxidized β-pinene or limonene products from SOA surfaces (Kamens et al., 1999), and a gas/particle partitioning absorption model has been found able to describe SOA yield well (Takekawa et al., 2003; Song et al., 2011)' (L224-228).

-Line 212, "number of organic molecules" - first I thought this was a mistake, but it seems the authors actually do mean the number of distinct molecules, i.e. the number of different elemental compositions measured (above some noise threshold). This might be good to specify explicitly here, to avoid misunderstandings (e.g. that "number" would mean "number concentration" or similar).

Response: We have changed it to 'formula number of assignable organic molecules in SOA' (L253).

-Line 231, "crack" has a very definite meaning in hydrocarbon chemistry, "fragment" is probably the word needed here.

Response: Thank you for the suggestion, we have changed it to 'fragment' and moved the sentence to L243.

-Line 249, "It seems" - could the authors speculate on the reason/mechanism of this inhibition? Could for example the OH produced by ozonolysis begin to cause more fragmentative oxidation at high O3 levels?

Response: Good suggestion, we have improved this sentence as follows: 'Thus, high concentration of ozone might have inhibitory effect on the formation of limonene SOA-associated organic compounds (e.g., due to ozonolysis-produced ·OH radicals begin to cause more oxidative fragmentation at high $O_3$ levels), whereas an enhancement effect on the formation of β-pinene SOA-associated organic compounds' (L278-280).

-Line 259, "overoxidation" - not a well-defined concept - I think I understand what is meant but the authors should still spell it out

Response: We have removed it to keep clarity.

-Lines 270-280, "dimer" often misspelled as "dimmer", please correct

Response: Thank you for pointing this out, we have corrected all the related typos for dimer in this manuscript.

-Line 273, "This may be due": this is almost certainly the explanation (already included in standard chemical mechanisms such as MCM) so the sentence could be a bit stronger here.

Response: We have changed to 'This is due to' (L308).

-Section 3.4, accretion reactions are probably going on BOTH in the gas phase and the particle phase (and potentially some gas-phase accretion products are, in parallel, broken up in the particle phase). So the measured particle-phase "dimers" may be quite different from the original gas-phase ones. (This is certainly implied already by the present discussion, but could be explicitly stated.)

Response: We have added 'Accretion reactions probably exist in both gas phase and particle phase, with partial gas phase accretion products partition into the particle phase and undergo further aging process. Thus, the measured particle-phase dimers may be quite different from the original gas-phase ones (Zhang et al., 2017)' (L392-394).

-RA_HOM in the conclusions is not defined, please spell out what is meant.

Response: RA_HOM is relative abundance of HOMs. We have changed it to 'The relative abundance of HOMs ($RA_{HOMs}$)' (L412).

-The last paragraph of the conclusions seems to be very general, and unrelated to the specific study performed here. While the presented arguments are correct, would it fit better in the discussion or even the introduction?

Response: The last paragraph of the conclusion is mainly intended to highlight that the complementary effects of different oxidants and environmental conditions on HOMs formation and evolution, which should also be considered when exploring the atmospheric processing of HOMs. We improved this paragraph as follows: 'In addition to ozone, other oxidants in the atmosphere may also impact the formation and evolution of HOMs. For example, $NO_2$ inhibited the formation of highly oxidized dimer products by suppression of autoxidation (Rissanen, 2018). When considerable amount of ·OH was present, the potential dimer sources can also be suppressed (Zhang et al., 2017). Moreover, Simon et al. found a continuously decreased oxidation level of α-pinene SOA and yields of HOMs as the temperature decreased from 25 to -50 °C (Simon et al., 2020). Similar result was also confirmed in urban field samples (Brean et al., 2020). Whether $O_3$ can exhibit a synergistic effect with other different oxidants and environmental conditions (e.g., humidity and temperature) in influencing HOMs formation and evolution is worthy to be explored in follow up studies. Current mass spectrometry techniques can only obtain information on the molecular formula of the HOMs, hindering the study of the detailed formation mechanism of many atmospheric precursors. Therefore, experimental conditions and new analytical techniques need to be developed to characterize HOMs rapidly and in detail (Bianchi et al., 2019)' (L422-431).

-Figure 8, WHAT are the markers colour-coded to? Please specify.

Response: The colour-codeds and the size of the dots represent the relative abundance on a logarithmic scale (L765).

-Figure 10, all the proposed channels are oxygen-increasing, and ALL of them proceed through the initial (well-established) Criegee intermediate => vinyl hydroperoxide => vinoxy radical + OH => peroxy radical sequence. I thus fail to understand why some of the channels are denoted "Criegee

channels" and "OIR" while others are not. (I would call the right-hand-side for example alkoxy channels - "hydroperoxy channel" is an acceptable name for the left-hand-side.) Also, wouldn't the alkoxy radical form equally well in reactions with HO2 or NO - or are these concentrations known to be low by the authors? Finally, the postulated H-shift from a C4 carbon in the (incorrectly labelled, see above) "Criegee channel" of b-pinene does not seem very plausible due to the associated ring strain - would not the H-shift from the alcohol carbon be the most likely one here also (as in the "hydroperoxide channel")?

Response: Thank you for your valuable suggestion, I have changed the "Criegee channel" to 'alkoxy channel' (L773). The term "OIR" refers to the H-shift $\rightarrow O_2 \rightarrow RO_2$ step in the alkoxy channel, so I have used "OIR" instead to avoid being cumbersome. In this study $NO_x$ was not introduced and the concentration of HOx radicals was not measured, which is also not the focus of this study. Finally the $C_{10}$ HOMs mechanism in Figure 10 has been revised.

The manuscript entitled "Large differences of highly oxygenated organic molecules (HOMs) and low volatile species in SOA formed from ozonolysis of β-pinene and limonene" reports chemical composition of SOA, particularly, HOM in particle-phase formed from ozonolysis of b-pinene and limonene. The SOA was formed in a flow tube with ~5 min reaction time. SOA composition was determined via filter collection followed by water extraction and analysis using Fourier transform ion cyclotron resonance mass spectrometry (FT-ICR-MS). This study investigated the effect of ozone concentration and compared the difference of the chemical composition between SOA formed in b-pinene and limonene ozonolysis. It was found that for b-pinene ozonolysis, as O3 concentration increased, particle size, OS, relative abundance (intensity-based) of HOM, HOM dimer abundance, and the fraction of LVOC in HOM increased; the relative abundance (intensity-based) of HOM monomer kept almost invariant. For SOA formed in limonene ozonolysis, as O3 increase, particle size increase, relative abundance of HOM stabilize, of HOM dimer, of HOM monomer remained generally stable; the fraction of LVOC OS, O/C, n(O), DBE, and MW first increased and then stabilized or slightly decreased with O3 increase. At the same O3 level, SOA formed in b-pinene ozonolysis had lower OS, MW, O/C, DBE, lower relative abundance of HOM, of HOM monomers, of HOM dimers, and more numbers of unique compounds (especially those with low OSc) than SOA form limonene ozonolysis.

This study addresses the chemical composition SOA determined on molecular level, which is an important and challenging topic of atmospheric chemistry. The manuscript fits the scope of ACP. I have the following comments for the authors to consider before publication.

Response: We thank the reviewer for the positive evaluation and valuable comments. The point-by-point responses to the comments are listed as below. The reviewer' comments are in black color and the author's responses are in blue color.

**General comments**

Some formulations of the manuscript are not easy to follow (e.g. lines 18-19, 82-83, 2s14-216, 247-248, 295-297, 305-306). And there are a number of grammar mistakes. I suggest the authors to polish the language throughout the manuscript.

Response: Thanks. We carefully polished the language throughout the manuscript.

**Specific comments**

Can the intensity-based abundance directly translate to concentration? (e.g. L220, Fig.3, and Fig.4). In another word, do all compounds have same sensitivity in MS so that peak intensity is directly proportional to the concentration?

Response: The FT-ICR MS instrument was used qualitatively rather than quantitatively. The reason

is that peak intensity of MS spectra is not proportional to their concentrations (e.g., due to the varied ionization efficiencies of different compounds). If the 'intensity-based abundance' is expressed in terms of concentration, it may be misinterpreted as quantitative result.

How much are the organic aerosol concentrations for the various $O_3$ levels? OA concentration can affect the partitioning of gas-phase species and thus interpretation of the dependence of chemical composition on $O_3$ concentration.

Response: The SOA mass concentrations is 0.2-1.5 µg m$^{-3}$. Yes, different ozone concentration will influence the ozonolysis rate of precursors. To clarify the effects of ozone concentration on SOA composition is exactly the aim of this study.

7 vs. Fig. 9, why is there no ULVOC in HOMs?

Response: This is mainly due the relative narrow definition of HOMs in this study. We have added a disclaimer in L210-212: 'It is noted that the current definition of HOMs does not count in HOM trimmers or other HOMs with higher oligomerization degrees, which is warranty to be explored in follow up studies'.

The foci of the abstract, conclusion, introduction is not exactly the same. (extracted using water). only WSOC?

Response: We have improved the abstract, conclusion, and introduction to keep consistency. Yes, the SOA filter samples were extracted with water, i.e., the WSOC were tested in this study.

L17, "5-13%" is not shown in the main text. How is this number obtained?

Response: We have changed to '14~20%', to keep it in consistency with the main text (L18).

L120, I suggest noting that the organic compounds are water soluble ones as only water is used for extraction.

Response: Thanks. We added 'To measure water-soluble organic compounds' in L137.

L192, why the standard of O/C<0.7 is used rather than nO<7 for the classification of HOMs (Bianchi et al., 2019)?

Response: For C10 organics, O/C < 0.7 and nO < 7 are equivalent. Other HOM monomers with C8 or C9 will have a larger O/C ratio while the number of O is 7.

L200, "which may be due to the formation of high molecular weight and low-volatile dimers" such a statement is not supported by an evidence. I suggest either omitting this or citing the figures on dimers in this study.

Response: We removed this sentence to avoid confusion.

L207-208, "…tends to increase oxygen reaction" is not clear.

Response: We have improved this sentence as follows: 'High concentration of ozone tends to convert less oxidized organic molecules to highly oxygenated organic molecules (HOMs) via oxygen-increasing-reactions on the carbon skeleton, i.e., addition of oxygen atom to the intermediate alkoxy radicals to form new alkoxy radicals (Kundu et al., 2012)' (L236-239).

L209, how the "abundance of organic peroxides" are obtained?

Response: The abundance of organic peroxides in β-pinene SOA is based on our previous study (Tong et al., 2018). The abundance of organic peroxide in limonene SOA is found from literature (Badali et al., 2015). We updated the main text and references as follows: 'The overall higher fraction of ULVOCs in β-pinene SOA than limonene SOA is in line with previously observed higher abundance of organic peroxides and aqueous radical yield of β-pinene SOA (Badali et al., 2015; Tong et al., 2016; Tong et al., 2018)' (L239-241).

L228-229, is it possible that dependence of composition on $O_3$ is related to the OH produced in ozonolysis as SOA formed via b-pinene ozonolysis likely does not contain C=C double bonds and thus can less likely react with $O_3$?

Response: We cannot fully agree with the point of 'SOA formed via β-pinene ozonolysis likely does not contain C=C double bonds'. Because the double bond equivalents of β-pinene SOA was found reach to 30, which means it contains substantial amounts of unsaturated compounds. The OH chemistry of SOA real can happen in this study. However, ozone was excessively injected into our flow tube. Thus, we do not suggest the OH chemistry to be the major causation to the SOA composition response to ozone concentrations.

L237, "as well as low H/C ratio organic molecules", I suggest citing Fig. 5 here. Otherwise, it is hard to follow.

Response: Thanks. We cited Figure 5 and improved this sentence and integrated it in L295-303.

L259, what does the overoxidation mean? Also by dissociation, fragmentation might be a better word.

Response: We have removed "the overoxidation and disassociation of preformed particulate HOMs may happen to limonene SOA" to avoid confusion.

L341, how the conclusion "β-pinene increases the possibility of carbonyl formation at high ozone concentrations" is not clear.

Response: We have removed "This suggests that β-pinene increases the possibility of carbonyl

formation at high ozone concentrations".

L358, why it is attributed to "the particle-phase chemistry" rather gas-phase reactions?

Response: We have changed to 'gas-phase reactions' (L396).

**Technical comments**

L24, "evolution mechanism of monoterpenes", do you mean evolution mechanism of monoterpene-derived SOA?

Response: We changed the "of" to 'for' (L26).

L67, "The more abundant atmospheric β-pinene and limonene" is not clear?

Response: We removed this sentence.

L205, this statement is for b-pinene.

Response: Thank you for pointing this out, we changed the starting sentence of this paragraph to 'The molecular weight (MW), O atom number, O/C ratio, double bound equivalents (DBE), carbon oxidation state ($OS_C$), and VOC subgroups of β-pinene SOA and limonene SOA were shown in Table 1' (L234-235).

L261, the last "or" should be "and".

Response: Thank you for pointing this out, we have changed to 'and' (L295).

L274, a "of" is missed before "dimmer".

Response: Thank you for pointing this out, we have added 'of' (L310).

L316, many or more?

Response: We improved the sentence as follows: 'At all three different ozone concentration conditions, β-pinene SOA always contained more low volatile organic compounds (LVOCs, ELVOCs, and ULVOCs) than limonene SOA, which distributed near the sugar alcohol ($C_nH_{2n+2}O_n$) line in Figure 9 (bule dashed line), indicating a greater impact of ozonolysis on β-pinene SOA's volatility diversity' (L351-354).

L370-371, I think that these sentences are not directly relevant to the main findings of this study.

Response: Agree. We have removed it.

L374, "with" is not correctly used here.

Response: We changed the sentence to 'with β-pinene to be found…… ' (L411).

Figure 8, color bar is missed

Response: Thank you for pointing this out, we have added color bar in the Figure 8.

**Response to the comments of Anonymous Referee #3**

The authors describe a set of experiments in which volatile organic compounds (VOCs), either limonene or beta-pinene, were oxidized by ozone in a flow tube. The subsequently formed aerosol particles were then analysed using different techniques. The main analytical tools used in the work are the scanning differential mobility particle sizer (SMPS) for measuring the size distribution of the formed aerosol particles, and Fourier transform ion cyclotron resonance mass spectrometry (FT-ICR MS) to characterize their chemical composition from filter samples. In its current form, I find that the manuscript lacks sufficient detail of the experimental procedure to support its conclusions. I feel that more data would be required to validate the results.

Response: We thank the reviewer for the critical and constructive comments. We tried our best to add more information on experimental details and data to support our conclusions. More details can be found from the following point by point response. The reviewer's comments are in black color and the author's responses are in blue color.

Also, I would strongly suggest making the data used for the manuscript freely available.

Response: All the dataset for this paper is available upon request from the corresponding authors (Dr. Haijie Tong: haijie.tong@hereon.de; Prof. Pingqing Fu: fupingqing@tju.edu.cn;). To facilitate the understanding of readers, we updated the Table S1 and added a new Figure S1 in SI. We also made the original SMPS data freely available now.

In addition, some of the introduction and conclusions sections are rather generic. For these, I would suggest for the authors to have a thorough look at the reference material, and form a coherent story based on that.

Response: Thanks. We thoroughly revised the introduction and conclusion sections.

Below, I have listed a set of concerns that I think need to be addressed before publication can be considered.

Response: We addressed your concerns point-by-point in the following.

- The reaction conditions. The authors list in Section 2.1 that they used a 7 l flow tube. This was operated with flows of 1 lpm $N_2$ to evaporate VOC (MFC 2 in Fig. 1) and around 2 lpm synthetic air through the ozone generator (MFC 3). Was MFC 1 used at all? The listed flows add up to 3 lpm: at this flow rate, the residence time in the tube would be 2 min 20 s, but the authors state a residence time of 5 min. Where does this discrepancy come from?

Response: MFC1, 2, and 3 were used for diluting/carrying gas (1 slpm $N_2$), gas for evaporating VOC (0.1 slpm $N_2$), and synthetic air (1.7 slpm), respectively. The residence time of all the gases will be around 2.5 minute. We updated this in the revised manuscript as follows: 'A flow of 1 bar and 0.1 standard liter per minute $N_2$ (99.999%, Westfalen AG) was used to evaporate the volatile

organic compounds (MFC2). Another 1 slpm $N_2$ flow was used as a diluting and carrier gas (MFC1) to introduce the gas phase precursors into the reactor for ~2.5 min gas-phase ozonolysis reaction. The $O_3$ was generated via passing synthetic air (Westfalen AG, 1.7 L min$^{-1}$) through MFC3 and a 185 nm UV light ($O_3$ generator, L.O.T.-Oriel GmbH & Co. KG)' (L116-120).

- In addition, a large fraction of the total flow consisted of nitrogen. However, HOM are formed in autoxidation, i.e., in the reaction with atmospheric $O_2$. Did you consider this effect?

Response: The effect of $O_2$ concentration on HOMs formation is not the interest of this study. However, it is worthy and warranty to investigate this point in our follow up studies.

- I suppose the listed ozone concentrations are without VOC in the flow tube. You have relatively imprecise estimates for the VOC concentrations. You could get better values by monitoring the ozone concentration change upon adding the VOC and calculating back from there. Did you do this?

Response: We did not monitor the ozone concentration change in this study, which is warranty to be measured in follow up studies.

- The reaction rate of limonene with ozone (2e-16 cm3 s-1) is over ten times that of bpinene with ozone (1.5e-17 cm3 s-1). So, for similar starting concentrations, you would, as a first approximation, expect over ten times more limonene to react. And so, assuming similar SOA yields, you would expect ten times more SOA from the limonene reaction. Limonene would be expected to form even more aerosol than this, as it typically has a higher SOA yield, and SOA yields typically increase with increasing reaction rate. Yet, you observe similar aerosol production, both in size distribution (Fig. 2) and in mass (Table S1). In mass, you even get clearly less mass for limonene for most conditions. How do you explain this?

Response: Figure S1 indicates that at 565 ppb $O_3$ condition, the peak concentration of limonene SOA (1484 μg m$^{-3}$ for 626 nm particles) is much higher than β-pinene SOA (1152 μg m$^{-3}$ for 684 nm particles), partially reflecting the higher SOA yield of limonene ozonolysis. The SOA concentration difference is not as high as ten times, which may be due to the exact concentration of limonene in the flow tube is lower than β-pinene. The evidence is that we observed substantial large particles (diameter > 600 nm) during the β-pinene SOA formation process (Figure S1a). The mass-size distribution of these particles does not change significantly and rapidly when the UV lights are turned off. We infer these large particles (Figure S1a) are β-pinene microdroplets produced during the precursor evaporation or due to condensation. In contrast, negligible number of big particles were observed during the limonene SOA formation process (Figure S1b). We added above discussion in L15-23 of SI.

- I'm very crudely estimating that in Fig. 2, you have around 15 000 particles per cubic centimetre for both limonene and b-pinene in the high ozone case. Assuming a mass mean diameter of 90 nm and a density of 1.5 g/cm3, this would correspond to a mass concentration of some 70 ug/m3. In Table S1, you collect around 200 ug of SOA in an hour. This would mean collecting around 3 m3 per hour, or 50 litres per minute. However, your total flow is much smaller. Am I missing something?

Or how do you get those numbers? Could you please provide the data for Fig. 2, or the whole SMPS data?

Response: The number- and mass-size distribution data were converted incorrectly from SMPS, and we apologize for this mistake. We corrected the data and updated the Figure 2 and Table S1. Therein the mass concentration data is based on a SOA density assumption of 1.0 g/cm$^3$. If we estimate the SOA density is 1.5 g/cm$^3$, the concentration of β-pinene and limonene SOA (particles < 600 nm) formed at 565 ppb O$_3$ condition will be around 700 and 900 µg m$^{-3}$, respectively. In this case, the theoretically collected aerosol mass for β-pinene and limonene SOA will be 115 µg and 148 µg. The exactly collected aerosol mass for β-pinene and limonene SOA were 205 µg and 245 µg. The difference between theoretical and experimental aerosol mass is 44% and 40% for β-pinene and limonene SOA, respectively.

**Additional point by point comments on the text:**

Intro, lines 38-65: the text flows poorly here, and is rather generic. I would reformulate your message and write it down again.

Response: We rewrote it as follows: 'Previous studies indicate that biogenic SOA comprise thousands of organic compounds, which exhibit a wide range of volatilities (Donahue et al., 2012; Ehn et al., 2014; Simon et al., 2020). Based on grouping the estimated effective saturation mass concentrations C$_0$ (Schervish and Donahue, 2020), the volatility of organic aerosols has been categorized into volatile organic compounds (VOCs, C$_0$ > 3×10$^6$ µg m$^{-3}$), intermediate volatile OC (IVOCs, 300 < C$_0$ < 3×10$^6$ µg m$^{-3}$), semivolatile OC (SVOCs, 0.3 < C$_0$ < 300 µg m$^{-3}$), low-volatile OC (LVOCs, 3×10$^{-5}$ < C$_0$ < 0.3 µg m$^{-3}$), extremely low-volatile OC (ELVOCs, 3×10$^{-9}$ < C$_0$ < 3×10$^{-5}$ µg m$^{-3}$), and ultralow-volatile organic compounds (ULVOCs, C$_0$ < 3×10$^{-9}$ µg m$^{-3}$), respectively (Hallquist et al., 2009; Simon et al., 2020). Iyer et al. suggested that after a single oxidant attack, BVOCs can be oxidized to low-volatility species on sub-second timescales, which consequently undergo decomposition or bear new particle formation (Iyer et al., 2021). Recent studies also showed that oxidation of BVOCs can produce large amounts of SOA particles via the nucleation of ULVOCs with the absence of sulfuric acid (Kirkby et al., 2016; Guo et al., 2022). As a result, the irreversible distribution of (extremely) low volatile oligomer on the aerosol surface is expected to be enhanced (Zhang et al., 2017). Beyond this, ELVOCs have been found playing a crucial role in the generation of atmospheric cloud condensation nuclei (Kerminen et al., 2012) and new particle formation in most continental regions (Jokinen et al., 2015). To disclose the role of different VOC subgroups in the formation and environmental impacts of SOA, it is of critical importance to chemically resolve their oxidation state and linkage with SOA evolution processes.

Highly oxygenated organic molecules (HOMs) have been found as a class of O-enriched multifunctional organic compounds (Tröstl et al., 2016). These HOMs frequently have a variety of redox functionalities (Zhang et al., 2017; Kirkby et al., 2016), playing an important role in the early growth of atmospheric organic aerosols (Ehn et al., 2014; Wang et al., 2020), and are closely associated with the formation of aqueous radicals (Tong et al., 2019). Ehn et al. found that HOMs in Hyytiälä's atmosphere, laboratory-generated α- and β-pinene SOA always have a O/C ≥ 0.7 (Ehn et al., 2012). Tröstl et al. suggested that α-pinene SOA-contained HOMs can be defined as C$_x$H$_y$O$_z$ with x = 8∼10, y = 12∼16 and z ≥ 6 for monomer and C$_x$H$_y$O$_z$ with x = 17∼20, y = 26∼32 and z ≥

8 for dimer (Tröstl et al., 2016). Tu et al. defined HOMs in fresh and aged biogenic (α-pinene, β-pinene, and limonene) SOA as assigned formulas having either $O/C \geq 0.6$ or carbon oxidation states $OS_C \geq 0$, which were also categorized into highly oxygenated and highly oxidized HOMs ($O/C \geq 0.6$ and $OS_C \geq 0$), highly oxygenated but less oxidized HOMs ($O/C \geq 0.6$ but $OS_C \geq 0$), and highly oxidized HOMs with a moderate level of oxygenation ($OS_C \geq 0$ but $H/C < 1.2$) for exploring the relative importance of oxygen content versus oxidation state (Kroll et al., 2011; Tu et al., 2016). Further study showed that monoterpene SOA-contained HOMs mainly composed of ELVOCs, LVOCs, and a small proportion of SVOCs (Li et al., 2019). Beyond the biogenic SOA, aromatic SOA and aged soot particles have also been found containing substantial fraction of HOMs (Molteni et al., 2018; Li et al., 2022). Respect to the formation mechanism of HOMs, autoxidation has been suggested to be an important pathway (Crounse et al., 2013; Rissanen et al., 2014). For instance, peroxyl radicals ($RO_2$) can undergo an intramolecular hydrogen atom shift (H-shift) to form a hydroperoxide functionality (HOO-) and an alkyl radical (RO), and then molecular oxygen rapidly attaches to form a new more oxidized $RO_2$ radical, and be repeated several times to form HOMs (Bianchi et al., 2019). Given the different yield, lifetime, and reactivity of HOMs in different types of SOA (Ehn et al., 2014; Jokinen et al., 2015; Pullinen et al., 2020; Shen et al., 2021; Guo et al., 2022), it is necessary to differentiate the compositional characteristics of HOMs in different types of SOA' (L60-94).

Line 60: the term ULVOC was not used by Bianchi et al., and has only been introduced later

Response: Thanks. We defined it in L65: ultralow-volatile OC (ULVOCs, $C_0 < 3 \times 10^{-9}$ μg m$^{-3}$).

Line 77: "opposite trends in SOA formation potential": what does this mean?

Response: We revised it as follows: 'Moreover, the oxidant (e.g., $O_3$ or ˙OH)-dependent SOA yield difference of β-pinene has a different magnitude from limonene (Jokinen et al., 2015; Mutzel et al., 2016)' (L57-58).

Line 80: "Nucleation rate of monoterpene SOA" does not really make sense here: nucleation rate and SOA yield are two separate (though connected) concepts. Here you seem to refer to SOA yield

Response: Thanks. We removed "Nucleation rate of monoterpene SOA " to avoid confusion.

Lines 82-83: I don't understand this sentence

Response: We have removed this sentence to avoid confusion.

Line 84-85: The same applies to low-volatility vapours. And the high ozone here means a high oxidation rate, and a high initial particle production rate. This makes condensation to particles a competitive sink for vapours, as opposed to wall loss.

Response: Thanks. We added 'and low-volatile' in L100.

Line 90: what does "gradient" mean here?

Response: We changed the "gradient" to 'different' to avoid misunderstanding (L104).

Line 102: what about the total flow, and MFC1? Already with the numbers here, 1/3 of the flow is $N_2$, which will already drop the $O_2$ concentration in the flow tube considerably. This may influence HOM formation, as they are formed in autoxidation, i.e., with atmospheric $O_2$.

Response: The total flow is 2.8 L/min. The flow rate through MFC1 is 1 L/min. The influence of oxygen concentration on the formation of HOMs is not the interest of this study. Even like this, we added the following disclaimer in L133-135: 'It is noted that dilution induced an oxygen concentration drop in the flow tube. The impacts of oxygen concentration on HOMs formation and evolution are out of the interest of this study and warranty to be explored in follow up studies.'

Line 103: What was the flow through MFC1? This is not listed, but with 1 lpm through the VOC and 2 lpm through the O3 generator, the total flow is already 3 lpm. In a 7 l flow tube, this should give a residence time of 2 min 20 s.

Response: The flow rate of $N_2$ through MFC1 is 1 L/min, of $N_2$ through MFC2 is 0.1 L/min, and of synthetic air through MFC3 is 1.7 L/min. Thus, the total air flow rates in the flow tube is 2.8 L/min. Therefore, the residence time of the flow in the flow tube is ~2.5 min. We corrected this in L117-120 now.

Line 105: how did you vary the ozone concentration?

Response: We varied the ozone concentration by changing the position of sleeve that used for shielding UV lamp.

Lines 107-108: at 1 ppm, the ozone lifetime towards limonene oxidation is around 200 s, and 2500 s for b-pinene. This means that in the 5 min almost all of the ozone should react in the limonene case, while in the beta-pinene case, only around 10 % reacts. Combined with the higher SOA yield of limonene, we should see much more SOA in the limonene case. This is, at least to some extent, seen in the SMPS data, but the opposite is seen in the collected mass. As the majority of the article is based on the filter analysis, this should be addressed.

Response: Figure S1 indicates that at 565 ppb $O_3$ condition, the peak concentration of limonene SOA (1484 µg m$^{-3}$ for 626 nm particles) is much higher than β-pinene SOA (1152 µg m$^{-3}$ for 684 nm particles), partially reflecting the higher SOA yield of limonene ozonolysis. The SOA concentration difference is not as high as ten times, which may be due to the exact concentration of limonene in the flow tube is lower than β-pinene. The evidence is that we observed substantial large particles (diameter > 600 nm) during the β-pinene SOA formation process (Figure S1a). The mass-size distribution of these particles does not change significantly and rapidly when the UV lights are turned off. We infer these large particles (Figure S1a) are β-pinene microdroplets produced during the precursor evaporation or due to condensation. In contrast, negligible number of big particles

were observed during the limonene SOA formation process (Figure S1b). We added above discussion in L15-23 of SI.

Line 115: what flow rate is this? To the SMPS?

Response: The flow rate is for the aerosol flow sucked by the pump.

Lines 116-117: what experimental tests? And what are these results on aqueous phase radicals that are mentioned a few times, but not really presented?

Response: We removed this redundant sentence.

Line 117: what wall loss was negligible? For sure there are different types of wall losses, both of vapours and of particles

Response: We removed the "wall loss" to avoid confusion.

Line 149: About the isotopic peaks: will this not lead to overestimation of some compounds, as isotopic signals from neighbouring masses overlap with them?

Response: The effects of isotope-induced overlapping is out of the interest of this study. We added the following discussions in L168-169: ', but more information on the FTICR-based analysis of isotopes in ambient aerosols can be found from our previous study (Xie et al., 2022)'.

Line 161: I don't really understand where this comes from. For example, C10H14O11 would fall under highly oxidized organic compounds, and I can't really think of any realistic examples of more oxidized HOM. Also, the method does not distinguish between –OOH group and two -OH groups, while the latter is twice as oxidized. Do you still think this classification makes sense?

Response: It comes from our recent work by Yun Zhang et al. (Zhang et al., 2021). The maximum Carbonyl Ratio (MCR) calculates the proportion of carbonyl groups that may contain the greatest amount of carbonyl as a means of estimating the maximum contribution to HOMs. We agree with you that MCR cannot differentiate the oxidation state of function groups with the same formula. We added the following discussions in L275-276: 'It is noted that MRC cannot differentiate the functional groups with the same formular, the improvement of which is warranty to be explored in follow up studies'.

Lines 189-190: The reference talks about ambient measurements, where volatilities were estimated with the SIMPOL parametrization. So, I would not use it for such a general statement.

Response: We agree. We revised this sentence as follows: 'Due to their low saturation vapor pressure, ambient HOMs (Vogel et al., 2016) and laboratory-generated HOMs (Jokinen et al., 2015; Roldin et al., 2019; Peräkylä et al., 2020) frequently comprise low-volatility organic compounds (LVOCs) even extremely low-volatility organic compounds (ELVOCs)' (L204-206).

Line 191: Do you mean that these compounds were assigned as HOM monomers and dimers?

Response: Yes. We improved the sentence as follows: formulae of $C_{8-10}H_{12-16}O_{6-9}$ and $C_{17-20}H_{26-32}O_{8-15}$ were assigned to HOM monomers and dimers formed from monoterpene ozonolysis (L206-207).

Lines 200-201: As the ozone increases, the whole reaction rate increases. And if SOA yield stays the same, or at least doesn't go dramatically down, the size and concentration of the particles should increase. So, this does not point to any specific compounds or mechanism yet

Response: Agree. We removed this sentence to avoid confusion.

Lines 201-202: Do you mean that organics promote the formation and growth, and thus increase the survival probability of the particles? Now it sounds like you mean that the specific organics themselves survive, which, as far as I know, is unknown

Response: We removed the "and survive to cloud condensation nuclei-active sizes" to avoid confusion.

Lines 207-208: "increase oxygen reaction": what does this mean?

Response: We clarified our discussion as follows: 'High concentration of ozone tends to convert less oxidized organic molecules to highly oxygenated organic molecules (HOMs) via oxygen-increasing-reactions on the carbon skeleton, i.e., addition of oxygen atom to the intermediate alkoxy radicals to form new alkoxy radicals (Kundu et al., 2012)' (L236-239).

Lines 209-210: Is this your result, or someone else's?

Response: It is from previous studies. We revised this sentence as follows: 'The overall higher fraction of ULVOCs in β-pinene SOA than limonene SOA is in line with previously observed higher abundance of organic peroxides and aqueous radical yield of β-pinene SOA (Badali et al., 2015; Tong et al., 2016; Tong et al., 2018)' (L239-241).

Line 214: I only see a slight decrease. Also, please avoid using "significantly" unless talking about statistical significance

Response: Good suggestion. We removed the "significantly" and revised the sentence as follows: 'but the formular number of limonene SOA gradually decreased' (L254-255).

Line 227: Abundance or relative abundance? I would expect the abundance of pretty much anything to go up with ozone in these conditions

Response: We corrected the "abundance" to 'relative abundance' (L265).

Line 237: "condensation reaction channel": what does this mean?

Response: We removed this term to avoid confusion.

Line 251: You are using quite a narrow definition of HOM, if I understand correctly (the predefined formulae on line 191). Looking at Fig. 5, I think many more would also qualify as HOM.

Response: Yes, we constrained the atomic number in this paper to avoid misassignment for HOMs. It is true that some HOMs may be filtered out due to the relative narrow definition. To complement the understanding, we added the following disclaimer in L210-212: 'It is noted that the current definition of HOMs does not count in HOM trimmers or other HOMs with higher oligomerization degrees, which is warranty to be explored in follow up studies'.

Line 251: We can't see from the plot that they are dimers. Also, it might be useful to label some of the largest peaks in the plot

Response: Agree. We removed the "as dimer".

Line 252: What is RA? Relative abundance? I don't think it's defined

Response: Yes, it is relative abundance, which is defined now in the line 282.

Lines 323-324: add references

Response: Thank you for pointing this out, we have added relevant references (L362).

Lines 329-330: The reference talks about slightly different element number ranges. Please be careful with references: now it sounds like Pospisilova et al would be talking about these exact numbers

Response: We removed the discussion on particle phase chemistry of HOMs to avoid misunderstanding.

Lines 330-332: Hard to follow sentence

Response: We removed it for clarity.

Line 336: High probability based on what?

Response: We revised this sentence and moved it to L365-366: 'There is also the probability of ozone first reacts with endocyclic double bond of limonene, which opens the chain to form alkoxy radical and then $C_{10}H_{14}O_7$'.

Line 337: where do these compounds come from here?

Response: They are shown in Figure 10.

Figure 10: is this all based on work of others? Or do you have some more support for these channels in particular? If not, I would leave this figure out

Response: The reaction pathways in Figure 10 are from literature studies. However, the molecular structure of reaction products is speculated by us.

Line 342: You are measuring monoterpene oxidation here, so presumably all of these are monoterpene oxidation products

Response: Yes. We revised this sentence for clarity as follows: 'a common monoterpene oxidation product $C_{10}H_{16}O_9$ has also been observed in β-pinene SOA in this study' (L369-370).

Line 343: Do you mean that C10H16O9 was only detected in b-pinene SOA?

Response: Yes, $C_{10}H_{16}O_9$ was only detected in b-pinene SOA in this study.

Lines 347-348: this is very speculative

Response: Agree. We revised it as follows: '$C_{17}H_{26}O_8$ may be produced by the decomposition of $C_{19}H_{28}O_{11}$ with a loss of ketene from the internally contained labile trioxide function group, and the conversion of the unstable acyl hydroperoxide groups to carboxyl groups (Kahnt et al., 2018)' (L380-382).

Lines 349-351: Hard to follow sentence. Also, C10H16O8 does not necessarily come from C10H15O8, it may also have other sources. And it only forms from C10H15O8 upon reaction with HO2. Finally, I don't think detecting these products "verifies" this pathway. At maximum, it does not contradict the pathway, but does not rule others out either. For instance, C20H30O12 could also form from C10H15O10 + C10H15O4.

Response: We removed this sentence to avoid misunderstanding.

Lines 355-358: First you talk about formation of oligomers from closed shell monomers. But the Berndt et al reference is for gas-phase formation of accretion products from RO2-RO2 reactions

Response: Agree. We revised it as follows: 'The gas-phase accretion reactions have been studied under laboratory conditions and were also suggested to play an important role in ambient environment (Berndt et al., 2018a)' (L394-395).

Lines 358-359: What observation? The reference is not about particle phase chemistry, but gas-phase

Response: Thank you for pointing this out, we have changed "particle-phase chemistry" to 'gas-phase reactions' (L396).

Line 363: I would also expect highly oxidized dimers in this range

Response: Dimers do exist from 450 to 650 Da, but trimers are the major molecules. We revised it as follows: 'Trimer-like compounds and highly oxidized dimers are typically in the range of 450~650 Da (Kundu et al., 2012)' (L401).

SI:
Table S1: These numbers don't make sense to me. At low O3, why do you get 70 ug of SOA in 0.4 h for beta-pinene, but only 30 ug in a full hour for limonene? And in nearly two hours, still only 30 ug? Limonene should have a higher SOA yield. Also, the SMPS shows more mass for limonene.

Response: This may be due to the plausible different aerosol volatility and sampling efficiency for β-pinene and limonene SOA.

Table S1: median-->medium

Response: We updated the Table S1.

Fig. S1: for each ozone concentration, are the results the average of the two filters?

Response: Yes. Figure 1 shows the averaged data.

**References:**

[revised manuscript text omitted]

---

## Author Response (AR2)

**Response to the comments of Anonymous Referee #1**

The authors have addressed many if not most of the issues I raised in my original review - but unfortunately the revision has introduced a number of new problematic issues. The overall readability of the manuscript is unfortunately still in places quite poor - especially the discussion of reaction mechanisms is extremely confused and confusing. (The introduction, in contrast, is substantially improved - thank you for that!) Substantial copy-editing, by someone who also understands the chemistry, is still needed before this can be published.

Response: We thank the reviewer for the positive evaluation to our last revision and the critical comments to the reaction mechanism discussion. Our point-by-point responses are shown in blue color as below.

Some major issues:

E.g. Line 239: "addition of oxygen atoms to the intermediate alkoxy radicals", this seems to imply some sort of $RO + O_3$ reactions going on, that would certainly be a major discovery for atmospheric science, but I see no evidence anywhere for anything like this. $O_3$ will attack the double bonds, but after that it plays no direct role in the oxidation - the repeated claims to the contrary would need much stronger evidence to be taken seriously. The observed $[O_3]$ dependent changes are very likely related to more indirect effects, including the effect of $O_3$ on $NO_x$ and $RO_2$ levels (and thus the fate of $RO_2$), the production of OH (this is correctly identified in the manuscript), and of course for limonene the probability of both double bonds being attacked (also correctly identified, though discussed in a confusing manner). The discussion on how $[O_3]$ affects the reaction mechanisms (in this section and elsewhere) must be thoroughly amended to reflect what is actually known about atmospheric oxidation.

Response: Thank you. To elucidate the term "oxygen-increasing-reactions (OIR)" clearly, we changed the sentence "addition of oxygen atoms to the intermediate alkoxy radicals" to "producing new alkyl radicals through $O_2$ addition to an existing alkyl radical → reaction between $RO_2$ radicals → isomerization of the alkoxy radicals" (L240-241). We agree with that $O_3$ can influence the $RO_2$ level and have added this point in Line 239. However, $NO_x$ is not used in this study. Thus, we do not suggest that indirect effect of $O_3$ on $NO_x$ level plays an important role in the observed $[O_3]$ dependent changes.

E.g. on line 267, the statement "accelerated autoxidation by higher concentrations of ozone" seems to convey (though I note this may just be a language issue rather than a chemistry issue) a profound ignorance about what autoxidation actually is: it is oxidation driven solely by $O_2$, where (by definition) no further oxidants are needed. Or in other words, $O_3$ only STARTS the autoxidation process, the $[O_3]$ concentration can by definition not accelerate (or decelerate) it. (Any effects must, as said above, be indirect, and relate e.g. to the fate of $RO_2$ as controlled by the concentrations of NO, other $RO_2$, etc).

Response: Agree. We have changed the "indicating their faster conversion to high oxygen-containing organic molecules via accelerated autooxidation by higher concentrations of ozone" to

"indicating the preferred formation of high oxygen-containing organic molecules at higher concentrations of ozone, which may be associated with the indirectly accelerated autoxidation via $O_3$-augmented $RO_2$ levels etc. " (L269-270).

Line 279: "oxidative fragmentation by OH", while OH certainly plays a big role here, this is a bit confusing: isn't it rather the $O_3$ - oxidation which tends to drive fragmentation? Or does this refer to fragmentation via acing-type reactions, i.e. after attacks by more than one oxidant (e.g. an initial $O_3$ attack, and then OH oxidation of the subsequent products)? Please clarify.

Response: Thanks. We have clarified this sentence by changing it to "fragmentation of first-generation products and condensing compounds (Hallquist et al., 2009; Kroll et al., 2007; Zhao et al., 2015) " (L283-284).

-Line 292: "generate OH via hydroperoxide channel", this is wrong or at least misleading, OH is generated in the Criegee step (common to BOTH channels shown in e.g. their figure 10). Also, while ozonolysis of the exocyclic bond will certainly make more sCI, also the sCI will typically form OH as even the thermal reaction is quite quick (bimolecular reactions of sCIs CAN happen, but are usually not the major sink). Thus this overall discussion is somewhat confused and misleading.

Response: Thank you. We removed the " via hydroperoxide channel" to avoid confusion.

Line 376, "Alternatively": this is not an "alternative", this is the mechanistic detailed explanation for exactly the same channel that the authors have just discussed. Again, this discussion is very confused, and seems to be repeating essentially similar things many times, without realising that they are talking about the same thing.

Response: Thanks. We improved this part as follows (L378-384):

"The accretion products formed from self-combination or cross-reaction of $RO_2$ radicals has been proposed to be generally important in producing higher-functionalized $RO_2$ radicals, HOM dimers, and SOA via the following pathways (Berndt et al., 2018b; Bianchi et al., 2019; Kahnt et al., 2018; Ehn et al., 2014; Berndt et al., 2018a; Tomaz et al., 2021) :

$RO_2 + RO_2 \rightarrow ROOR + O_2$           (6)

$RO_2 + R'O_2 \rightarrow RO_4R' \rightarrow RO\cdots O_2\cdots OR' \rightarrow ROOR' + O_2$   (7)

$RO_2 + R'O \rightarrow RO_3R'$              (8)

Where the $RO\cdots O_2\cdots R'O$ means a cage structure intermediate formed from the asymmetric cleavage of tetroxide and eventually converts to ROOR' (Lee et al., 2016)".

Line 383: The Shi et al reference is unfortunately very unlikely to explain anything here, at least not in the gas phase: the rate coefficients corresponding to the energetics shown in that paper for

the RO2 + alkene reactions are far too slow to yield any products (in the gas phase) at any reasonable concentrations (as can easily be verified e.g. using simple transition state theory as an upper limit to the rates). Note: the RO2 + alkene energetics in that paper are likely more-or-less correct, it's just the conclusions that are not compatible even with their own numbers. And the single-reference methods employed therein cannot even begin to describe the actual RO2 + R'O2 reaction mechanism, so for those their barriers are of course completely off (actual multireference calculations, cited even by themselves, show that they are compatible with atmospheric observations). RO2 + alkene might be playing a role in the liquid phase (along with a number of other better-known condensation reactions), but certainly not in the gas phase.

Response: Thanks. We have removed the discussions on reaction of $RO_2$ with alkene to avoid misleading.

Selected minor issues:

-use of the word "saturated" in the abstract: this is an extremely unfortunate word choice, as "saturation" can mean several different things even in this exact context (from the number of double bonds to the vapor pressure, to the levelling off that the authors apparently refer to). Please formulate more clearly what is meant here (presumably the levelling of with respect to the ozone concentration, or something similar). I realise that the authors may have picked the term "saturate" from my original comment - I apologise for that!

Response: Thank you. We changed "saturate" to "exhibit higher yield" (L20-21).

-Line 67: "bear" new particles - bear is not needed here ("undergo" is a decent verb also for NPF).

Response: Thanks. We have removed the "bear" to avoid redundance (L68).

-Line 76: "redox functionalities": this doesn't mean anything. Reformulate

Response: Thank you. We have deleted this term to avoid confusion (L77).

-Line 242: "redox activity of SOA": meaningless

Response: We have changed "redox activity" to "the oxidative potentials" (L244).

-Line 249: "non-condensation reaction": what is meant by this?

Response: Thanks. We clarified this term by changing it to "non-condensation reactions that dominated by hemi-acetal reactions followed by hydrperoxide and Criegee radical reactions might play an important role in the limonene SOA formation" (L251).

-Line 254: "formula number of assignable organic molecules": I don't understand what is meant by this.

Response: We changed the term "assignable" to "identified" for clarification (L256).

Line 373, "oxidation degree of RO2 termination": what is meant by this?

Response: We changed this sentence to "reflecting the higher oxidation degree of HOMs in β-pinene SOA than limonene SOA-associated HOMs" for clarification (L375-376).

**Response to the comments of Anonymous Referee #2**

The authors have addressed most of my comments with few exceptions detailed below. I recommend publishing the manuscript after addressing these minor comments.

Response: We thank the reviewer for the positive evaluation to our last revision. Our point-by-point responses are shown in blue color as below.

1. Initial comment #1, it is worth adding a note that the peak intensity is not directly translatable to abundance/concentration.

Response: Thanks. We have added "It is noted that peak intensity of MS spectrum is not directly translatable to abundance or concentration" in L173-174.

2. Initial comment #7, for a C9-compound, if it contains 6 oxygen atoms, it would be classified as non-HOM (O/C<0.7), which is slightly different from the common classification of HOM definition (Bianchi, Kurten et al. 2019). While it is totally fine to have a different definition, I suggest noting the difference so that readers can better follow.

Response: Thank you. We clarified this point in the line 212-213: "It is noted that the current definition of HOMs is different from previous studies and does not count in HOM trimmers or other HOMs with higher oligomerization degrees, which is warranty to be explored in follow up studies".

3. Initial comment #11, even though the double bond equivalents of β-pinene SOA are high, they may contain only rings and C=O and do not necessarily contain C=C double bonds. As the C=C of β-pinene react with ozone in the first step, how are "new" C=C bonds formed? In another word, what is the potential chemical mechanism behind "the conversion of less oxidized organic molecules into high oxygen-containing organic molecules" promoted by excessive ozone?

Response: We agree with that high double bond equivalents of β-pinene SOA does not means high C=C abundance. We suggest that oxygen-increasing-reactions play important role in the conversion of less oxidized organic molecules into high oxygen-containing organic molecules. To clarify our discussion point, we have revised the initial lines 228-229 as follows: "indicating the preferred formation of high oxygen-containing organic molecules at higher concentrations of ozone, which may be associated with the indirectly accelerated autoxidation via $O_3$-augmented $RO_2$ levels etc." (L269-270).

**Reference**

Berndt, T., Mentler, B., Scholz, W., Fischer, L., Herrmann, H., Kulmala, M., and Hansel, A.: Accretion product formation from ozonolysis and OH radical reaction of α-pinene: Mechanistic insight and the influence of isoprene and ethylene, Environ. Sci. Technol., 52, 11069-11077, https://doi.org/10.1021/acs.est.8b02210, 2018a.

Berndt, T., Scholz, W., Mentler, B., Fischer, L., Herrmann, H., Kulmala, M., and Hansel, A.: Accretion product formation from self- and cross-reactions of $RO_2$ radicals in the atmosphere, Angew. Chem. Int. Ed., 57, 3820-3824, https://doi.org/10.1002/anie.201710989, 2018b.

Bianchi, F., Kurten, T., Riva, M., Mohr, C., Rissanen, M. P., Roldin, P., Berndt, T., Crounse, J. D., Wennberg, P. O., Mentel, T. F., Wildt, J., Junninen, H., Jokinen, T., Kulmala, M., Worsnop, D. R., Thornton, J. A., Donahue, N., Kjaergaard, H. G., and Ehn, M.: Highly oxygenated organic molecules (HOM) from gas-phase autoxidation involving peroxy radicals: A key contributor to atmospheric aerosol, Chem. Rev., 119, 3472-3509, https://doi.org/10.1021/acs.chemrev.8b00395, 2019.

Ehn, M., Thornton, J. A., Kleist, E., Sipila, M., Junninen, H., Pullinen, I., Springer, M., Rubach, F., Tillmann, R., Lee, B., Lopez-Hilfiker, F., Andres, S., Acir, I. H., Rissanen, M., Jokinen, T., Schobesberger, S., Kangasluoma, J., Kontkanen, J., Nieminen, T., Kurten, T., Nielsen, L. B., Jorgensen, S., Kjaergaard, H. G., Canagaratna, M., Maso, M. D., Berndt, T., Petaja, T., Wahner, A., Kerminen, V. M., Kulmala, M., Worsnop, D. R., Wildt, J., and Mentel, T. F.: A large source of low-volatility secondary organic aerosol, Nature, 506, 476-479, https://doi.org/10.1038/nature13032, 2014.

Kahnt, A., Vermeylen, R., Iinuma, Y., Safi Shalamzari, M., Maenhaut, W., and Claeys, M.: High-molecular-weight esters in α-pinene ozonolysis secondary organic aerosol: structural characterization and mechanistic proposal for their formation from highly oxygenated molecules, Atmos. Chem. Phys., 18, 8453-8467, https://doi.org/10.5194/acp-18-8453-2018, 2018.

Lee, R., Gryn'ova, G., Ingold, K. U., and Coote, M. L.: Why are sec-alkylperoxyl bimolecular self-reactions orders of magnitude faster than the analogous reactions of tert-alkylperoxyls? The unanticipated role of CH hydrogen bond donation, Phys. Chem. Chem. Phys., 18, 23673-23679, https://doi.org/10.1039/c6cp04670c, 2016.

Tomaz, S., Wang, D., Zabalegui, N., Li, D., Lamkaddam, H., Bachmeier, F., Vogel, A., Monge, M. E., Perrier, S., Baltensperger, U., George, C., Rissanen, M., Ehn, M., El Haddad, I., and Riva, M.: Structures and reactivity of peroxy radicals and dimeric products revealed by online tandem mass spectrometry, Nat. Commun., 12, 300, https://doi.org/10.1038/s41467-020-20532-2, 2021.